# ON THE DECISION BOUNDARIES OF DEEP NEURAL NETWORKS: A TROPICAL GEOMETRY PERSPECTIVE

## ABSTRACT

This work tackles the problem of characterizing and understanding the decision boundaries of neural networks with piecewise linear non-linearity activations. We use tropical geometry, a new development in the area of algebraic geometry, to provide a characterization of the decision boundaries of a simple neural network of the form (Affine, ReLU, Affine). Specifically, we show that the decision boundaries are a subset of a tropical hypersurface, which is intimately related to a polytope formed by the convex hull of two zonotopes. The generators of the zonotopes are precise functions of the neural network parameters. We utilize this geometric characterization to shed lights on new perspectives of three tasks. In doing so, we propose a new tropical perspective for the lottery ticket hypothesis, where we see the effect of different initializations on the tropical geometric representation of the decision boundaries. Also, we leverage this characterization as a new set of tropical regularizers, which deal directly with the decision boundaries of a network. We investigate the use of these regularizers in neural network pruning (removing network parameters that do not contribute to the tropical geometric representation of the decision boundaries) and in generating adversarial input attacks (with input perturbations explicitly perturbing the decision boundaries geometry to change the network prediction of the input).

## 1 INTRODUCTION

Deep Neural Networks (DNNs) have demonstrated outstanding performance across several research domains, including computer vision (Krizhevsky et al., 2012), speech recognition (Hinton et al., 2012), natural language processing (Bahdanau et al., 2015; Devlin et al., 2018), quantum chemistry (Schütt et al., 2017), and healthcare (Ardila et al., 2019; Zhou et al., 2019) to name a few (LeCun et al., 2015). Nevertheless, a rigorous interpretation of their success remains evasive (Shalev-Shwartz & Ben-David, 2014). For instance, and in an attempt to uncover the expressive power of DNNs, Montufar et al. (2014) studied the complexity of functions computable by DNNs that have piecewise linear activations. They derived a lower bound on the maximum number of linear regions. Several other works have followed to improve such estimates under certain assumptions (Arora et al., 2018). In addition, and in attempt to understand some of the subtle behaviours DNNs exhibit, *e.g.* the sensitive reaction of DNNs to small input perturbations, several works directly investigated the decision boundaries induced by a DNN used for classification. The work of Seyed-Mohsen Moosavi-Dezfooli (2019) showed that the smoothness of these decision boundaries and their curvature can play a vital role in network robustness. Moreover, He et al. (2018a) studied the expressiveness of these decision boundaries at perturbed inputs and showed that these boundaries do not resemble the boundaries around benign inputs. Li et al. (2018) showed that under certain assumptions, the decision boundaries of the last fully connected layer of DNNs will converge to a linear SVM. Also, Beise et al. (2018) showed that the decision regions of DNNs with width smaller than the input dimension are unbounded.

More recently, and due to the popularity of the piecewise linear ReLU as an activation function, there has been a surge in the number of works that study this class of DNNs in particular. As a result, this has incited significant interest in new mathematical tools that help analyze piecewise linear functions, such as tropical geometry. While tropical geometry has shown its potential in many applications such as dynamic programming (Joswig & Schröter, 2019), linear programming (Allamigeon et al., 2015), multi-objective discrete optimization (Joswig & Loho, 2019), enumerative geometry (Mikhalkin, 2004), economics (Akian et al., 2009; Mai Tran & Yu, 2015), it has only been

recently used to analyze DNNs. For instance, Zhang et al. (2018) showed an equivalency between the family of DNNs with piecewise linear activations and integer weight matrices and the family of tropical rational maps, *i.e.* ratio between two multi-variate polynomials in tropical algebra. The work of Zhang et al. (2018) was mostly concerned about characterizing the complexity of a DNN and specifically counting the number of linear regions, into which the function represented by the DNN can divide the input space, by counting the number of vertices of some polytope representation. This novel approach recovered the results of Montufar et al. (2014) with a much simpler analysis.

**Contributions.** In this paper, we take the results of Zhang et al. (2018) some steps further and present a novel perspective on the decision boundaries of DNNs using tropical geometry. To that end, our contributions are three-fold. **(i)** We derive a geometric representation (convex hull between two zonotopes) for a super set to the decision boundaries of a DNN in the form (Affine, ReLU, Affine). **(ii)** We demonstrate support for the lottery ticket hypothesis (Frankle & Carbin, 2019) using a geometric perspective. **(iii)** We leverage the geometrical representation of the decision boundaries (the decision boundaries polytope) in two interesting applications: network purning and adversarial attacks. In regards to *tropical pruning*, we provide a new geometric perspective in which one can directly compress the decision boundaries polytope efficiently resulting in only minor perturbations to the decision boundaries. We conduct extensive experiments on AlexNet (Krizhevsky et al., 2012) and VGG16 (Simonyan & Zisserman, 2014) on SVHN (Netzer et al., 2011), CIFAR10, and CIFAR 100 (Krizhevsky & Hinton, 2009) datasets, in which $90\%$ pruning rate can be achieved with a marginal drop in testing accuracy. As for *tropical adversarial attack*, we show that one can construct input adversaries that can change network predictions by perturbing the decision boundaries polytope. We conduct extensive experiments on MNIST (LeCun, 1998).

## 2 PRELIMINARIES TO TROPICAL GEOMETRY

We provide here some preliminaries to tropical geometry. For a thorough detailed review, we refer the reader to the work of Itenberg et al. (2009); Maclagan & Sturmfels (2015).

**Definition 1.** *(Tropical Semiring[1]) The tropical semiring $\mathbb{T}$ is the triplet $\{\mathbb{R} \cup \{-\infty\}, \oplus, \odot\}$, where $\oplus$ and $\odot$ define tropical addition and tropical multiplication, respectively. They are denoted as:*

$$x \oplus y = \max\{x, y\}, \qquad x \odot y = x + y, \qquad \forall x, y \in \mathbb{T}.$$

*It can be readily shown that $-\infty$ is the additive identity and $0$ is the multiplicative identity.*

Given the previous definition, a tropical power can be formulated as $x^{\odot a} = x \odot x \cdots \odot x = a.x$, for $x \in \mathbb{T}$, $a \in \mathbb{N}$, where $a.x$ is standard multiplication. Moreover, the tropical quotient can be defined as: $x \oslash y = x - y$ where $x - y$ is the standard subtraction. For ease of notation, we write $x^{\odot a}$ as $x^a$. Now, we are in a position to define tropical polynomials, their solution sets and tropical rationals.

**Definition 2.** *(Tropical Polynomials) For $\mathbf{x} \in \mathbb{T}^d$, $c_i \in \mathbb{R}$ and $\mathbf{a}_i \in \mathbb{N}^d$, a d-variable tropical polynomial with $n$ monomials. $f: \mathbb{T}^d \to \mathbb{T}^d$ can be expressed as:*

$$f(\mathbf{x}) = (c_1 \odot \mathbf{x}^{\mathbf{a}_1}) \oplus (c_2 \odot \mathbf{x}^{\mathbf{a}_2}) \oplus \cdots \oplus (c_n \odot \mathbf{x}^{\mathbf{a}_n}), \ \forall \ \mathbf{a}_i \neq \mathbf{a}_j \ \text{when} \ i \neq j.$$

*We use the more compact vector notation $\mathbf{x}^{\mathbf{a}} = x_1^{a_1} \odot x_2^{a_2} \cdots \odot x_d^{a_d}$ where $\mathbf{x}, \mathbf{a} \in \mathbb{R}^d$. Moreover and for ease of notation, we will denote $c_i \odot \mathbf{x}^{\mathbf{a}_i}$ as $c_i \mathbf{x}^{\mathbf{a}_i}$ throughout the paper.*

**Definition 3.** *(Tropical Rational Functions) A tropical rational function is a standard difference or equivalently, a tropical quotient of two tropical polynomials: $f(\mathbf{x}) - g(\mathbf{x}) = f(\mathbf{x}) \oslash g(\mathbf{x})$.*

Algebraic curves or hypersurfaces in algebraic geometry, which are the solution sets to polynomials, can be analogously extended to tropical polynomials too.

**Definition 4.** *(Tropical Hypersurfaces) A tropical hypersurface of a tropical polynomial $f(\mathbf{x}) = c_1 \mathbf{x}^{\mathbf{a}_1} \oplus \cdots \oplus c_n \mathbf{x}^{\mathbf{a}_n}$ is the set of points $\mathbf{x}$ where $f$ is attained by two or more monomials in $f$, i.e.*

$$\mathcal{T}(f) := \{\mathbf{x} \in \mathbb{R}^d : c_i \mathbf{x}^{\mathbf{a}_i} = c_j \mathbf{x}^{\mathbf{a}_j} = f(\mathbf{x}), \ \text{for some} \ \mathbf{a}_i \neq \mathbf{a}_j\}.$$

Tropical hypersurfaces divide the domain of $f$ into convex regions, where $f$ is linear in each region. Moreover, every tropical polynomial can be associated with a Newton polytope.

---

[1] A semiring is a ring that lacks an additive inverse.

**Definition 5.** *(Newton Polytopes) The Newton polytope of a tropical polynomial* $f(\mathbf{x}) = c_1\mathbf{x}^{\mathbf{a}_1} \oplus \cdots \oplus c_n\mathbf{x}^{\mathbf{a}_n}$ *is the convex hull of the exponents* $\mathbf{a}_i \in \mathbb{N}^d$ *regarded as points in* $\mathbb{R}^d$, *i.e.*

$$\Delta(f) := ConvHull\{\mathbf{a}_i \in \mathbb{R}^d : i = 1, 2, \ldots, n \text{ and } c_i \neq -\infty\}.$$

A tropical polynomial determines a dual subdivision, which can thus be constructed by projecting the collection of upper faces (UF) in $\mathcal{P}(f) := \text{ConvHull}\{(\mathbf{a}_i, c_i) \in \mathbb{R}^d \times \mathbb{R} : i = 1, \ldots, n\}$ to $\mathbb{R}^d$. That is to say, the dual subdivision determined by $f$ is given as $\delta(f) := \{\pi(p) \subset \mathbb{R}^d : p \in \text{UF}(\mathcal{P}(f))\}$ where $\pi : \mathbb{R}^d \times \mathbb{R} \rightarrow \mathbb{R}^d$ is the projection that drops the last coordinate. It has been shown by Maclagan & Sturmfels (2015) that the tropical hypersurface $\mathcal{T}(f)$ is the (d-1)-skeleton of the polyhedral complex dual to $\delta(f)$. So, each vertex of $\delta(f)$ corresponds to one region in $\mathbb{R}^d$ where $f$ is linear. Zhang et al. (2018) showed an equivalency between tropical rational maps and any neural network $f : \mathbb{R}^n \rightarrow \mathbb{R}^k$ with piecewise linear activations and integer weights through the following theorem.

**Theorem 1.** *(Tropical Characterization of Neural Networks, Zhang et al. (2018)). A feedforward neural network with integer weights and real biases with piecewise linear activation functions is a function* $f : \mathbb{R}^n \rightarrow \mathbb{R}^k$, *whose coordinates are tropical rational functions of the input, i.e.,*

$$f(\mathbf{x}) = H(\mathbf{x}) \oslash Q(\mathbf{x}) = H(\mathbf{x}) - Q(\mathbf{x}),$$

*where $H$ and $Q$ are tropical polynomials.*

While this result is new in the context of tropical geometry, it is not surprising, since any piecewise linear function can be represented as a difference of two max functions over a set of hyperplanes Melzer (1986). Mathematically, that is to say if $f$ is a piecewise linear function, it can be written as $f(\mathbf{x}) = \max_{i \in [m]}\{\mathbf{a}_i^\top\mathbf{x}\} - \max_{j \in [n]}\{\mathbf{b}_j^\top\mathbf{x}\}$, where $[m] = \{1, \ldots, m\}$ and $[n] = \{1, \ldots, n\}$. Thus, it is clear that each of the two maxima above is a tropical polynomial recovering Theorem 1.

## 3 Decision Boundaries of Deep Neural Networks as Polytopes

In this section, we analyze the decision boundaries of a network in the form (Affine, ReLU, Affine) using tropical geometry. For ease, we use ReLUs as the non-linear activation, but any other piecewise linear function can also be used. The functional form of this network is: $f(\mathbf{x}) = \mathbf{B}\max(\mathbf{A}\mathbf{x} + \mathbf{c}_1, \mathbf{0}) + \mathbf{c}_2$, where $\max(.)$ is an element-wise operator. The outputs of the network $f$ are the logit scores. Throughout this section, we assume[2] that $\mathbf{A} \in \mathbb{Z}^{p \times n}$, $\mathbf{B} \in \mathbb{Z}^{2 \times p}$, $\mathbf{c}_1 \in \mathbb{R}^p$ and $\mathbf{c}_2 \in \mathbb{R}^2$. For ease of notation, we only consider networks with two outputs, *i.e.* $\mathbf{B}^{2 \times p}$, where the extension to a multi-class output follows naturally and it is discussed in the **appendix**. Now, since $f$ is a piecewise linear function, each output can be expressed as a tropical rational as per Theorem 1. If $f_1$ and $f_2$ refer to the first and second outputs respectively, we have $f_1(\mathbf{x}) = H_1(\mathbf{x}) \oslash Q_1(\mathbf{x})$ and $f_2(\mathbf{x}) = H_2(\mathbf{x}) \oslash Q_2(\mathbf{x})$, where $H_1, H_2, Q_1$ and $Q_2$ are tropical polynomials. In what follows and for ease of presentation, we present our main results where the network $f$ has no biases, *i.e.* $\mathbf{c}_1 = \mathbf{0}$ and $\mathbf{c}_2 = \mathbf{0}$, and we leave the generalization to the **appendix**.

**Theorem 2.** *For a bias-free neural network in the form of* $f(\mathbf{x}) : \mathbb{R}^n \rightarrow \mathbb{R}^2$ *where* $\mathbf{A} \in \mathbb{Z}^{p \times n}$ *and* $\mathbf{B} \in \mathbb{Z}^{2 \times p}$, *let* $R(\mathbf{x}) = H_1(\mathbf{x}) \odot Q_2(\mathbf{x}) \oplus H_2(\mathbf{x}) \odot Q_1(\mathbf{x})$ *be a tropical polynomial. Then:*

- *Let* $\mathcal{B} = \{\mathbf{x} \in \mathbb{R}^n : f_1(\mathbf{x}) = f_2(\mathbf{x})\}$ *defines the decision boundaries of $f$, then* $\mathcal{B} \subseteq \mathcal{T}(R(\mathbf{x}))$.

- $\delta(R(\mathbf{x})) = ConvHull(\mathcal{Z}_{\mathbf{G}_1}, \mathcal{Z}_{\mathbf{G}_2})$. $\mathcal{Z}_{\mathbf{G}_1}$ *is a zonotope in* $\mathbb{R}^n$ *with line segments* $\{(\mathbf{B}^+(1, j) + \mathbf{B}^-(2, j))[\mathbf{A}^+(j, :), \mathbf{A}^-(j, :)]\}_{j=1}^p$ *and shift* $(\mathbf{B}^-(1, :) + \mathbf{B}^+(2, :))\mathbf{A}^-$. $\mathcal{Z}_{\mathbf{G}_2}$ *is a zonotope in* $\mathbb{R}^n$ *with line segments* $\{(\mathbf{B}^-(1, j) + \mathbf{B}^+(2, j))[\mathbf{A}^+(j, :), \mathbf{A}^-(j, :)]\}_{j=1}^p$ *and shift* $(\mathbf{B}^+(1, :) + \mathbf{B}^-(2, :))\mathbf{A}^-$. *Note that* $\mathbf{A}^+ = \max(\mathbf{A}, 0)$ *and* $\mathbf{A}^- = \max(-\mathbf{A}, 0)$. *The line segment* $(\mathbf{B}^+(1, j) + \mathbf{B}^-(2, j))[\mathbf{A}^+(j, :), \mathbf{A}^-(j, :)]$ *has end points* $\mathbf{A}^+(j, :)$ *and* $\mathbf{A}^-(j, :)$ *in* $\mathbb{R}^n$ *and scaled by* $(\mathbf{B}^+(1, j) + \mathbf{B}^-(2, j))$.

The proof for Theorem 2 is left for the **appendix**.

**Digesting Theorem 2.** Theorem 2 can be broken into two major results. The first, which is on the algebra side, *i.e.* finding the solution set to tropical polynomials, states that the decision boundaries

---

[2]Without loss of generality, as one can very well approximate real weights as fractions by multiplying by least common multiple of the denominators as discussed in Zhang et al. (2018).

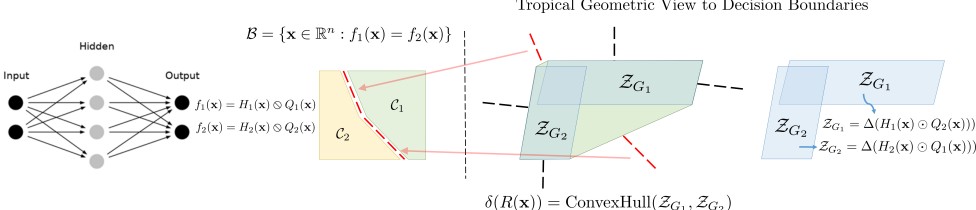

Figure 1: **Decision Boundaries as Geometric Structures.** The decision boundaries $\mathcal{B}$ (in red) comprise two linear pieces separating classes $\mathcal{C}_1$ and $\mathcal{C}_2$. As per Theorem 2, the dual subdivision of this single hidden neural network is the convex hull between the zonotopes $\mathcal{Z}_{\mathbf{G}_1}$ and $\mathcal{Z}_{\mathbf{G}_2}$. The normals to the dual subdivison $\delta(R(\mathbf{x}))$ are in one-to-one correspondence to the tropical hypersurface $\mathcal{T}(R(\mathbf{x}))$, which is a superset to the decision boundaries $\mathcal{B}$. Note that some of the normals to $\delta(R(\mathbf{x}))$ (in red) are parallel to the decision boundaries.

$\mathcal{B}$ is a subset of the tropical hypersurface of the tropical polynomial $R(\mathbf{x})$, *i.e.* $\mathcal{T}(R(\mathbf{x}))$. The second result, which is on the geometry side, of Theorem 2 relates the tropical polynomial $R(\mathbf{x})$ to the geometric representation of the solution set to $R(\mathbf{x})$, *i.e.* $\mathcal{T}(R(\mathbf{x}))$, referred to as the dual subdivision, *i.e.* $\delta(R(\mathbf{x}))$. In particular, Theorem 2 states that the dual subdivision for a network $f$ is the convex hull of two zonotopes denoted as $\mathcal{Z}_{G_1}$ and $\mathcal{Z}_{G_2}$. Note that this dual subdivision is a function of only the network parameters $\mathbf{A}$ and $\mathbf{B}$.

Theorem 2 bridges the gap between the behaviour of the decision boundaries $\mathcal{B}$, through the super-set $\mathcal{T}(R(\mathbf{x}))$, and the polytope $\delta(R(\mathbf{x}))$, which is the convex hull of two zonotopes. It is worthwhile to mention that Zhang et al. (2018) discussed a special case of the first part of Theorem 2 for a neural network with a single output and a score function $s(\mathbf{x})$ to classify the output. To the best of our knowledge, this work is the first to propose a tropical geometric formulation of a super-set containing the decision boundaries of a multi-class classification neural network. In particular, the first result of Theorem 2 states that one can alter the network, *e.g.* by pruning network parameters, while preserving the decision boundaries $\mathcal{B}$, if one preserves the tropical hypersurface of $R(\mathbf{x})$ or $\mathcal{T}(R(\mathbf{x}))$. While preserving the tropical hypersurfaces can be equally difficult to preserving the decision boundaries directly, the second result of Theorem 2 comes in handy. For a bias free network, $\pi$ becomes an identity mapping with $\delta(R(\mathbf{x})) = \Delta(R(\mathbf{x}))$, and thus the dual subdivision $\delta(R(\mathbf{x}))$, which is the Newton polytope $\Delta(R(\mathbf{x}))$ in this case, becomes a well structured geometric object that can be exploited to preserve decision boundaries. Since Maclagan & Sturmfels (2015) (Proposition 3.1.6) showed that the tropical hypersurface is the skeleton of the dual to $\delta(R(\mathbf{x}))$, the normal lines to the edges of the polytope $\delta(R(\mathbf{x}))$ are in one-to-one correspondence with the tropical hypersurface $\mathcal{T}(R(\mathbf{x}))$. Figure 1 details this intimate relation between the decision boundaries, tropical hypersurface $\mathcal{T}(R(\mathbf{x}))$, and normals to $\delta(R(\mathbf{x}))$. Before any further discussion, we recap the definition of zonotopes.

**Definition 6.** *Let* $\mathbf{u}^1, \ldots, \mathbf{u}^p \in \mathbb{R}^n$. *The zonotope formed by* $\mathbf{u}^1, \ldots, \mathbf{u}^p$ *is defined as* $\mathcal{Z}(\mathbf{u}^1, \ldots, \mathbf{u}^p) := \{\sum_{i=1}^{p} x_i \mathbf{u}^i : 0 \leq x_i \leq 1\}$. *Equivalently, the zonotope can be expressed with respect to the generator matrix* $\mathbf{U} \in \mathbb{R}^{p \times n}$, *where* $\mathbf{U}(i, :) = \mathbf{u}^{i\top}$ *as* $\mathcal{Z}_{\mathbf{U}} := \{\mathbf{U}^\top \mathbf{x} : \forall \mathbf{x} \in [0, 1]^p\}$.

Another common definition for zonotopes is the Minkowski sum (refer to appendix A for the definition of the Minkowski sum) of a set of line segments that start from the origin with end points $\mathbf{u}^1, \ldots, \mathbf{u}^p \in \mathbb{R}^n$. It is also well known that the number of vertices of a zonotope is polynomial in the number of line segments. That is to say, $|\text{vert}(\mathcal{Z}_{\mathbf{U}})| \leq 2 \sum_{i=0}^{n-1} \binom{p-1}{i} = \mathcal{O}(p^{n-1})$ (Gritzmann & Sturmfels, 1993).

While Theorem 2 presents a strong relation between a polytope (convex hull of two zonotopes) and the decision boundaries, it remains unclear how such a polytope can be efficiently constructed. Although the number of vertices of a zonotope is polynomial in the number of its generating line segments, fast algorithms for enumerating these vertices are still restricted to zonotopes with line segments starting at the origin (Stinson et al., 2016). Since the line segments generating the zonotopes in Theorem 2 have arbitrary end points, we present the next result that transforms these line segments into a generator matrix of line segments starting from the origin, as prescribed in Definition 6. This result is essential for the efficient computation of the zonotopes in Theorem 2.

**Proposition 1.** *Consider* $p$ *line segments in* $\mathbb{R}^n$ *with two arbitrary end points as follows* $\{[\mathbf{u}_1^i, \mathbf{u}_2^i]\}_{i=1}^p$. *The zonotope formed by these line segments is equivalent to the zonotope formed by the line segments* $\{[\mathbf{u}_1^i - \mathbf{u}_2^i, \mathbf{0}]\}_{i=1}^p$ *with a shift of* $\sum_{i=1}^p \mathbf{u}_2^i$.

Figure 2: **Effect of Different Initializations on the Decision Boundaries Polytope.** From left to right: training dataset, decision boundaries polytope of original network followed by the decision boundaries polytope during several iterations of pruning with different initializations.

The proof is left for the **appendix**. As per Proposition 1, the generator matrices of zonotopes $\mathcal{Z}_{\mathbf{G}_1}, \mathcal{Z}_{\mathbf{G}_2}$ in Theorem 2 can be defined as $\mathbf{G}_1 = \text{Diag}[(\mathbf{B}^+(1,:)) + (\mathbf{B}^-(2,:))]\mathbf{A}$ and $\mathbf{G}_2 = \text{Diag}[(\mathbf{B}^+(2,:)) + (\mathbf{B}^-(1,:))]\mathbf{A}$, both with shift $(\mathbf{B}^-(1,:) + \mathbf{B}^+(2,:) + \mathbf{B}^+(1,:) + \mathbf{B}^-(2,:))\mathbf{A}^-$.

In what follows, we show several applications for Theorem 2. We begin by leveraging the geometric structure to help in reaffirming the behaviour of the lottery ticket hypothesis.

## 4    TROPICAL VIEW TO THE LOTTERY TICKET HYPOTHESIS

The lottery ticket hypothesis was recently proposed by Frankle & Carbin (2019), in which the authors surmise the existence of sparse trainable sub-networks of dense, randomly-initialized, feedforward networks that—when trained in isolation—perform as well as the original network in a similar number of iterations. To find such sub-networks, Frankle & Carbin (2019) propose the following simple algorithm: perform standard network pruning, initialize the pruned network with the same initialization that was used in the original training setting, and train with the same number of epochs. They hypothesize that this should result in a smaller network with a similar accuracy to the larger dense network. In other words, a subnetwork can have similar decision boundaries to the original network. While in this section we do not provide a theoretical reason for why this proposed pruning algorithm performs favorably, we utilize the geometric structure that arises from Theorem 2 to reaffirm such behaviour. In particular, we show that the orientation of the decision boundaries polytope $\delta(R(\mathbf{x}))$, known to be a superset to the decision boundaries $\mathcal{T}(R(\mathbf{x}))$, is preserved after pruning with the proposed initialization algorithm of Frankle & Carbin (2019). On the other hand, pruning routines with a different initialization at each pruning iteration will result in a severe variation in the orientation of the decision boundaries polytope. This leads to a large change in the orientation of the decision boundaries, which tends to hinder accuracy.

To this end, we train a neural network with 2 inputs ($n = 2$), 2 outputs, and a single hidden layer with 40 nodes ($p = 40$). We then prune the network by removing the smallest $x\%$ of the weights. The pruned network is then trained using different initializations: (i) the same initialization as the original network (Frankle & Carbin, 2019), (ii) Xavier (Glorot & Bengio, 2010), (iii) standard Gaussian and (iv) zero mean Gaussian with variance of 0.1. Figure 2 shows the evolution of the decision boundaries polytope, *i.e.* $\delta(R(\mathbf{x}))$, as we perform more pruning (increasing the $x\%$) with different initializations. It is to be observed that the orientation of the polytopes $\delta(R(\mathbf{x}))$ vary much more for all different initialization schemes as compared to the lottery ticket initialization. This gives an indication that lottery ticket initialization indeed preserves the decision boundaries throughout the evolution of pruning. Another approach to investigate the lottery ticket could be by observing the polytopes representing the functional form of the network directly, *i.e.* $\delta(H_{\{1,2\}}(\mathbf{x}))$ and $\delta(Q_{\{1,2\}}(\mathbf{x}))$, in lieu of the decision boundaries polytopes. However, this does not provide conclusive answers to the lottery ticket, since there can exist multiple functional forms, and correspondingly multiple polytopes $\delta(H_{\{1,2\}}(\mathbf{x}))$ and $\delta(Q_{\{1,2\}}(\mathbf{x}))$, for networks with the same decision boundaries. This is why we explicitly focus our analysis on $\delta(R(\mathbf{x}))$, which is directly related to the decision boundaries of the network. Further discussions and experiments are left for the **appendix**.

## 5    TROPICAL NETWORK PRUNING

Network pruning has been identified as an effective approach for reducing the computational cost and memory usage during network inference time. While pruning dates back to the work of LeCun et al. (1990) and Hassibi & Stork (1993), it has recently gained more attention. This is due to the fact that most neural networks over-parameterize commonly used datasets. In network pruning, the task is to find a smaller subset of the network parameters, such that the resulting smaller network has similar decision boundaries (and thus supposedly similar accuracy) to the original over-parameterized network. In this section, we show a new geometric approach towards network pruning. In particu-

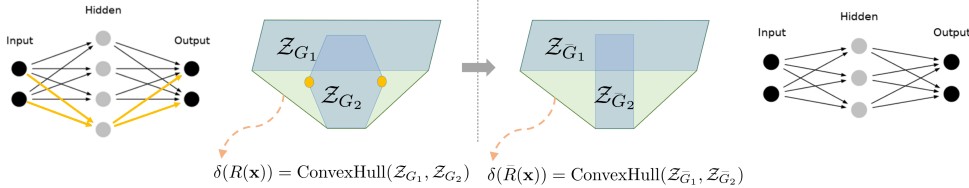

$$\delta(R(\mathbf{x})) = \text{ConvexHull}(\mathcal{Z}_{G_1}, \mathcal{Z}_{G_2}) \qquad \delta(\bar{R}(\mathbf{x})) = \text{ConvexHull}(\mathcal{Z}_{\bar{G}_1}, \mathcal{Z}_{\bar{G}_2})$$

Figure 3: **Tropical Pruning Pipeline.** Pruning the $4^{th}$ node, or equivalently removing the two yellow vertices of zonotope $\mathcal{Z}_{G_2}$ does not affect the decision boundaries polytope which will not lead to any change in accuracy.

lar, as indicated by Theorem 2, preserving the polytope $\delta(R(\mathbf{x}))$ preserves a superset to the decision boundaries $\mathcal{T}(R(\mathbf{x}))$, and thus supposedly the decision boundaries themselves.

**Motivational Insight.** For a single hidden layer neural network, the dual subdivision to the decision boundaries is the polytope that is the convex hull of two zonotopes, where each is formed by taking the Minkowski sum of line segments (Theorem 2). Figure 3 shows an example where pruning a neuron in the neural network has no effect on the dual subdivision polytope and equivalently no effect on the accuracy, since the decision boundaries of both networks remain the same.

**Problem Formulation.** Given the motivational insight, a natural question arises: *Given an over-parameterized binary neural network* $f(\mathbf{x}) = \mathbf{B} \max(\mathbf{A}\mathbf{x}, \mathbf{0})$, *can one construct a new neural network, parameterized by some sparser weight matrices* $\tilde{\mathbf{A}}$ *and* $\tilde{\mathbf{B}}$, *such that this smaller network has a dual subdivision* $\delta(\tilde{R}(\mathbf{x}))$ *that preserves the decision boundaries of the original network?*

In order to address this question, we propose the following general optimization problem

$$\min_{\tilde{\mathbf{A}}, \tilde{\mathbf{B}}} d\Big(\delta(\tilde{R}(\mathbf{x})), \delta(R(\mathbf{x}))\Big) = \min_{\tilde{\mathbf{A}}, \tilde{\mathbf{B}}} d\Big(\text{ConvexHull}\left(\mathcal{Z}_{\tilde{\mathbf{G}}_1}, \mathcal{Z}_{\tilde{\mathbf{G}}_2}\right), \text{ConvexHull}\left(\mathcal{Z}_{\mathbf{G}_1}, \mathcal{Z}_{\mathbf{G}_2}\right)\Big). \quad (1)$$

The function $d(.)$ defines a distance between two geometric objects. Since the generators $\tilde{\mathbf{G}}_1$ and $\tilde{\mathbf{G}}_2$ are functions of $\tilde{\mathbf{A}}$ and $\tilde{\mathbf{B}}$ (as per Theorem 2), this optimization problem can be challenging to solve. However, for pruning purposes, one can observe from Theorem 2 that if the generators $\tilde{\mathbf{G}}_1$ and $\tilde{\mathbf{G}}_2$ had fewer number of line segments (rows), this corresponds to a fewer number of rows in the weight matrix $\tilde{\mathbf{A}}$ (sparser weights). To this end, we observe that if $\tilde{\mathbf{G}}_1 \approx \mathbf{G}_1$ and $\tilde{\mathbf{G}}_2 \approx \mathbf{G}_2$, then $\delta(\tilde{R}(\mathbf{x})) \approx \delta(R(\mathbf{x}))$, and thus the decision boundaries tend to be preserved as a consequence. Therefore, we propose the following optimization problem as a surrogate to Problem (1)

$$\min_{\tilde{\mathbf{A}}, \tilde{\mathbf{B}}} \tfrac{1}{2}\Big(\left\|\tilde{\mathbf{G}}_1 - \mathbf{G}_1\right\|_F^2 + \left\|\tilde{\mathbf{G}}_2 - \mathbf{G}_2\right\|_F^2\Big) + \lambda_1 \left\|\tilde{\mathbf{G}}_1\right\|_{2,1} + \lambda_2 \left\|\tilde{\mathbf{G}}_2\right\|_{2,1}. \quad (2)$$

The matrix mixed norm for $\mathbf{C} \in \mathbb{R}^{n \times k}$ is defined as $\|\mathbf{C}\|_{2,1} = \sum_{i=1}^{n} \|\mathbf{C}(i,:)\|_2$, which encourages the matrix $\mathbf{C}$ to be row sparse, *i.e.* complete rows of $\mathbf{C}$ are zero. Note that $\tilde{\mathbf{G}}_1 = \text{Diag}[\text{ReLU}(\tilde{\mathbf{B}}(1,:)) + \text{ReLU}(-\tilde{\mathbf{B}}(2,:))]\tilde{\mathbf{A}}, \tilde{\mathbf{G}}_2 = \text{Diag}[\text{ReLU}(\tilde{\mathbf{B}}(2,:)) + \text{ReLU}(-\tilde{\mathbf{B}}(1,:))]\tilde{\mathbf{A}}$, and $\text{Diag}(\mathbf{v})$ rearranges the elements of vector $\mathbf{v}$ in a diagonal matrix. We solve the aforementioned problem with alternating optimization over the variables $\tilde{\mathbf{A}}$ and $\tilde{\mathbf{B}}$, where each sub-problem is solved in closed form. Details of the optimization and the extension to multi-class case are left for the **appendix**.

*Extension to Deeper Networks.* For deeper networks, one can still apply the aforementioned optimization for consecutive blocks. In particular, we prune each consecutive block of the form (Affine,ReLU,Affine) starting from the input and ending at the output of the network.

**Experiments on Tropical Pruning.** Here, we evaluate the performance of the proposed pruning approach as compared to several classical approaches on several architectures and datasets. In particular, we compare our tropical pruning approach against Class Blind (CB), Class Uniform (CU) and Class Distribution (CD) Han et al. (2015); See et al. (2016). In Class Blind, all the parameters across all nodes of a layer are sorted by magnitude where $x\%$ with smallest magnitudes are pruned. Similar to Class Blind, Class Uniform prunes the parameters with smallest $x\%$ magnitudes per node in a layer as opposed to sorting all parameters in all nodes as in Class Blind. Lastly, Class Distribution performs pruning of all parameters for each node in the layer, just as in Class Uniform, but the parameters are pruned based on the standard deviation $\sigma_c$ of the magnitude of the parameters per node. Since fully connected layers in deep neural networks tend to have much higher memory complexity than convolutional layers, we restrict our focus to pruning fully connected layers. We

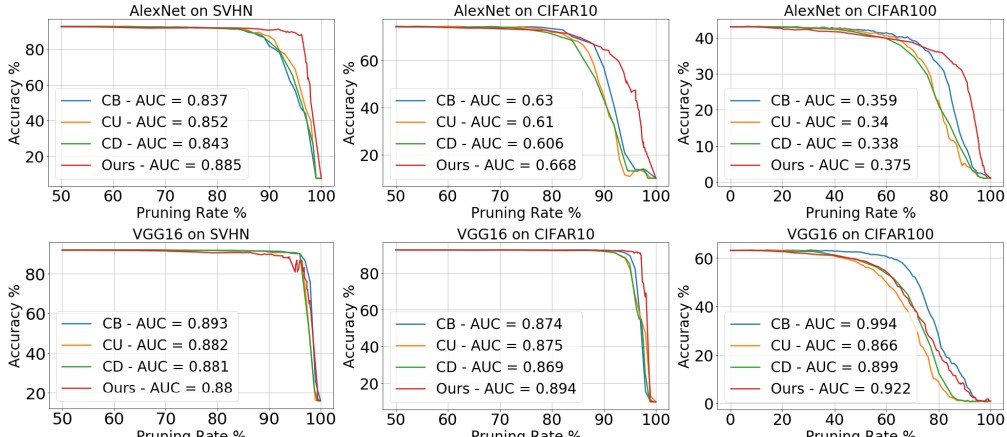

Figure 4: **Results of Tropical Pruning.** Pruning-accuracy plots for AlexNet (top) and VGG16 (bottom) trained on SVHN, CIFAR10, and CIFAR100, pruned with our tropical method and three other pruning methods.

train AlexNet and VGG16 on SVHN , CIFAR10, and CIFAR 100 datasets. We observe that we can prune more than 90% of the classifier parameters for both networks without affecting the accuracy. Moreover, we can boost the pruning ratio using our method without affecting the accuracy by simply retraining the network biases only.

*Setup.* We adapt the architectures of AlexNet and VGG16, since they were originally trained on ImageNet (Deng et al., 2009), to account for the discrepancy in the input resolution. The fully connected layers of AlexNet and VGG16 have sizes of (256,512,10) and (512,512,10), respectively on SVHN and CIFAR100 with the last layer replaced to 100 for CIFAR100. All networks were trained to baseline test accuracy of (92%,74%,43%) for AlexNet on SVHN, CIFAR10 and CIFAR100, respectively and (92%,92%,70%) for VGG16. To evaluate the performance of pruning, following previous works (Han et al., 2015), we report the area under the curve (AUC) of the pruning-accuracy plot. The higher the AUC is, the better the trade-off is between pruning rate and accuracy. For efficiency purposes, we run the optimization in Problem (2) for a single alternating iteration to identify the rows in $\tilde{\mathbf{A}}$ and elements of $\tilde{\mathbf{B}}$ that will be pruned, since an exact pruning solution might not be necessary. The algorithm and the parameters setup to solving (2) is left for the **appendix**.

*Results.* Figure 4 shows the pruning comparison between our tropical approach and the three aforementioned popular pruning schemes on both AlexNet and VGG16 over the different datasets. Our proposed approach can indeed prune out as much as 90% of the parameters of the classifier without sacrificing much of the accuracy. For AlexNet, we achieve much better performance in pruning as compared to other methods. In particular, we are better in AUC by 3%, 3%, and 2% over other pruning methods on SVHN, CIFAR10 and CIFAR100, respectively. This indicates that the decision boundaries can indeed be preserved by preserving the dual subdivision polytope. For VGG16, we perform similarly well on both SVHN and CIFAR10 and slightly worse on CIFAR100. While the performance achieved here is comparable to the other pruning schemes, if not better, we emphasize that our contribution does not lie in outperforming state-of-the-art pruning methods, but rather in giving a new geometry based perspective to network pruning. We conduct more experiments, where only the biases of the network or the biases of the classifier are fine tuned after pruning . Retraining biases can be sufficient as they do not contribute to the orientation of the decision boundaries polytope, thereafter the decision boundaries, but only a translation. Discussion on biases and more results are left for the **appendix**.

## 6 TROPICAL ADVERSARIAL ATTACKS

DNNs are notoriously known to be susceptible to adversarial attacks. In fact, adding small imperceptible noise, referred to as adversarial attacks, at the input of these networks can hinder their performance. Several works investigated the decision boundaries of neural networks in the presence of adversarial attacks. For instance, Khoury & Hadfield-Menell (2018) analyzed high dimensional geometry of adversarial examples by the means of manifold reconstruction. Also, He et al. (2018b) crafted adversarial attacks by estimating the distance to the decision boundaries using random search directions. In this work, we provide a tropical geometric view to this problem. where we show how Theorem 2 can be leveraged to construct a tropical geometric based targeted adversarial attack.

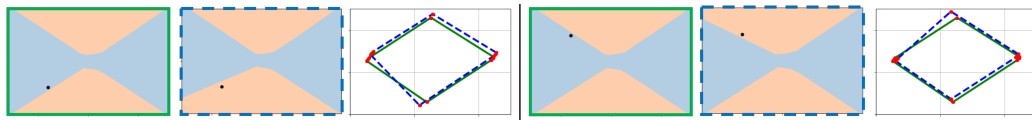

Figure 5: **Dual View of Tropical Adversarial Attacks.** We show the effects of tropical adversarial attacks on a synthetic binary dataset at two different input points (in black). From left to right: the decision regions of the original and perturbed models, and decision boundaries polytopes (green for original and blue for perturbed).

**Dual View to Adversarial Attacks.** For a classifier $f : \mathbb{R}^n \to \mathbb{R}^k$ and input $\mathbf{x}_0$ that is classified as $c$, a standard formulation for targeted adversarial attacks flips the classifier prediction to a particular class $t$ and it is usually defined as follows

$$\min_\eta \quad \mathcal{D}(\eta) \quad \text{s.t.} \quad \arg\max_i \ f_i(\mathbf{x}_0 + \eta) = t \neq c. \tag{3}$$

This objective aims at computing the lowest energy input noise $\eta$ (measured by $\mathcal{D}$) such that the the new sample $(\mathbf{x}_0 + \eta)$ crosses the decision boundaries of $f$ to a new classification region. Here, we present a dual view to adversarial attacks. Instead of designing a sample noise $\eta$ such that $(\mathbf{x}_0 + \eta)$ belongs to a new decision region, one can instead fix $\mathbf{x}_0$ and perturb the network parameters to move the decision boundaries in a way that $\mathbf{x}_0$ appears in a new classification region. In particular, let $\mathbf{A}_1$ be the first linear layer of $f$, such that $f(\mathbf{x}_0) = g(\mathbf{A}_1 \mathbf{x}_0)$. One can now perturb $\mathbf{A}_1$ to alter the decision boundaries and relate the perturbation to the input perturbation as follows

$$g((\mathbf{A}_1 + \xi_{\mathbf{A}_1})\mathbf{x}_0) = g\left(\mathbf{A}_1 \mathbf{x}_0 + \xi_{\mathbf{A}_1} \mathbf{x}_0\right) = g(\mathbf{A}_1 \mathbf{x}_0 + \mathbf{A}_1 \eta) = f(\mathbf{x}_0 + \eta). \tag{4}$$

From this dual view, we observe that traditional adversarial attacks are intimately related to perturbing the parameters of the first linear layer through the linear system: $\mathbf{A}_1 \eta = \xi_{\mathbf{A}_1} \mathbf{x}_0$. To this end, Theorem 2 provides explicit means to geometrically construct adversarial attacks by means of perturbing decision boundaries. In particular, since the normals to the dual subdivision polytope $\delta(R(\mathbf{x}))$ of a given neural network represent the tropical hypersurface set $\mathcal{T}(R(\mathbf{x}))$ which is, as per Theorem 2, a superset to the decision boundaries set $\mathcal{B}$, $\xi_{\mathbf{A}_1}$ can be designed to result in a minimal perturbation to the dual subdivision that is sufficient to change the network prediction of $\mathbf{x}_0$ to the targeted class $t$. Based on this observation, we formulate the problem as follows

$$\begin{aligned}
\min_{\eta, \xi_{\mathbf{A}_1}} \quad & \mathcal{D}_1(\eta) + \mathcal{D}_2(\xi_{\mathbf{A}_1}) \\
\text{s.t.} \quad & -\text{loss}(g(\mathbf{A}_1(\mathbf{x}_0 + \eta)), t) \leq -1; \quad -\text{loss}(g(\mathbf{A}_1 + \xi_{\mathbf{A}_1})\mathbf{x}_0, t) \leq -1; \\
& (\mathbf{x}_0 + \eta) \in [0,1]^n, \quad \|\eta\|_\infty \leq \epsilon_1; \quad \|\xi_{\mathbf{A}_1}\|_{\infty,\infty} \leq \epsilon_2; \quad \mathbf{A}_1 \eta - \xi_{\mathbf{A}_1} \mathbf{x}_0 = 0.
\end{aligned} \tag{5}$$

The loss is the standard cross-entropy loss. The first row of constraints ensures that the network prediction is the desired target class $t$ when the input $\mathbf{x}_0$ is perturbed by $\eta$, and equivalently by perturbing the first linear layer $\mathbf{A}_1$ by $\xi_{\mathbf{A}_1}$. This is identical to $f_1$ as proposed by Carlini & Wagner (2016). Moreover, the third and fourth constraints guarantee that the perturbed input is feasible and that the perturbation is bounded, respectively. The fifth constraint is to limit the maximum perturbation on the first linear layer, while the last constraint enforces the dual equivalence between input perturbation and parameter perturbation. The function $\mathcal{D}_2$ captures the perturbation of the dual subdivision polytope upon perturbing the first linear layer by $\xi_{\mathbf{A}_1}$. For a single hidden layer neural network parameterized as $(\mathbf{A}_1 + \xi_{\mathbf{A}_1}) \in \mathbb{R}^{p \times n}$ and $\mathbf{B} \in \mathbb{R}^{2 \times p}$ for the $1^{st}$ and $2^{nd}$ layers respectively, $\mathcal{D}_2$ can capture the perturbations in each of the two zonotopes discussed in Theorem 2.

$$\mathcal{D}_2(\xi_{\mathbf{A}_1}) = \tfrac{1}{2} \sum_{j=1}^{2} \left\| \text{Diag}\left(\mathbf{B}^+(j,:)\right)\xi_{\mathbf{A}_1} \right\|_F^2 + \left\| \text{Diag}\left(\mathbf{B}^-(j,:)\right)\xi_{\mathbf{A}_1} \right\|_F^2. \tag{6}$$

The derivation, discussion, and extension of (6) to multi-class neural networks is left for the **appendix**. We solve Problem (5) with a penalty method on the linear equality constraints, $\mathbf{A}_1 \eta = \xi_{\mathbf{A}_1} \mathbf{x}_0$, where each penalty step is solved with ADMM (Boyd et al., 2011) in a similar fashion to the work of Xu et al. (2018). The details of the algorithm are left for the **appendix**.

**Motivational Insight to the Dual View.** This intuition is presented in Figure 5. We train a single hidden layer neural network where the size of the input is 2 with 50 hidden nodes and 2 outputs on a simple dataset as shown in Figure 5. We then solve Problem 5 for a given $\mathbf{x}_0$ shown in black. We show the decision boundaries for the network with and without the perturbation at the first linear layer $\xi_{\mathbf{A}_1}$. Figure 5 shows that indeed perturbing an edge of the dual subdivision polytope, by perturbing the first linear layer, corresponds to perturbing the decision boundaries and results in miss-classifying $\mathbf{x}_0$. Interestingly and as expected, perturbing different decision boundaries corresponds to perturbing different edges of the dual subdivision. In particular, one can see from Figure 5

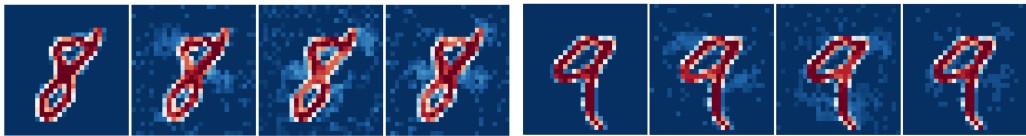

Figure 6: **Effect of Tropical Adversarial Attacks on MNIST Dataset.** We show qualitative examples of adversarial attacks, produced by solving Problem (5), on two digits (8,9) from MNIST. From left to right, images are classified as [8,7,5,4] and [9,7,5,4] respectively.

that altering the decision boundaries, by altering the dual subdivision polytope through perturbations in the first linear layer, can result in miss-classifying a previously correctly classified input $\mathbf{x}_0$.

*MNIST Experiment.* Here, we design perturbations to misclassify MNIST images. Figure 7 shows several adversarial examples that change the network prediction for digits 8 and 9 to digits 7, 5, and 4, respectively. In some cases, the perturbation $\eta$ is as small as $\epsilon = 0.1$, where $\mathbf{x}_0 \in [0, 1]^n$. Several other adversarial results are left for the **appendix**. We again emphasize that our approach is not meant to be compared with (or beat) state of the art adversarial attacks, but rather to provide a novel geometrically inspired perspective that can shed new light in this field.

## 7 CONCLUSION

In this paper, we leverage tropical geometry to characterize the decision boundaries of neural networks in the form (Affine, ReLU, Affine) and relate it to well-studied geometric objects such as zonotopes and polytopes. We leaverage this representation in providing a tropical perspective to support the lottery ticket hypothesis, network pruning and designing adversarial attacks. One natural extension for this work is a compact derivation for the characterization of the decision boundaries of convolutional neural networks (CNNs) and graphical convolutional networks (GCNs).

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

# A  PRELIMINARIES AND DEFINITIONS.

**Fact 1.** $P\tilde{+}Q = \{p + q, \forall p \in P \text{ and } q \in Q\}$ *is the Minkowski sum between two sets $P$ and $Q$.*

**Fact 2.** *Let $f$ be a tropical polynomial and let $a \in \mathbb{N}$. Then*

$$\mathcal{P}(f^a) = a\mathcal{P}(f).$$

Let both $f$ and $g$ be tropical polynomials, Then

**Fact 3.**

$$\mathcal{P}(f \odot g) = \mathcal{P}(f)\tilde{+}\mathcal{P}(g). \tag{7}$$

**Fact 4.**

$$\mathcal{P}(f \oplus g) = ConvexHull\Big(\mathcal{V}\left(\mathcal{P}(g)\right) \cup \mathcal{V}\left(\mathcal{P}(g)\right)\Big). \tag{8}$$

Note that $\mathcal{V}(\mathcal{P}(f))$ is the set of vertices of the polytope $\mathcal{P}(f)$.

## B PROOF OF THEOREM 2

**Theorem 3.** *For a bias-free neural network in the form of $f(\mathbf{x}) : \mathbb{R}^n \to \mathbb{R}^2$ where $\mathbf{A} \in \mathbb{Z}^{p \times n}$ and $\mathbf{B} \in \mathbb{Z}^{2 \times p}$, and let $R(\mathbf{x}) = H_1(\mathbf{x}) \odot Q_2(\mathbf{x}) \oplus H_2(\mathbf{x}) \odot Q_1(\mathbf{x})$ be a tropical polynomial, then*

- *If the decision boundaries of $f$ is given by the set $\mathcal{B} = \{x \in \mathbb{R}^n : f_1(\mathbf{x}) = f_2(\mathbf{x})\}$, then we have $\mathcal{B} \subseteq \mathcal{T}(R(\mathbf{x}))$.*

- $\delta(R(\mathbf{x})) = ConvHull(\mathcal{Z}_{\mathbf{G}_1}, \mathcal{Z}_{\mathbf{G}_2})$ *where $\mathcal{Z}_{\mathbf{G}_1}$ is a zonotope in $\mathbb{R}^n$ with line segments $\{(\mathbf{B}(1,j)^+ + \mathbf{B}(2,j)^-)[\mathbf{A}(j,:)^+, \mathbf{A}(j,:)^-]_{j=1}^p\}$ with shift $(\mathbf{B}(1,:)^- + \mathbf{B}(2,:)^+)\mathbf{A}^-$ while $\mathcal{Z}_{\mathbf{G}_2}$ is a zonotope in $\mathbb{R}^n$ with line segments $\{(\mathbf{B}(1,j)^- + \mathbf{B}(2,j)^+)[\mathbf{A}(j,:)^+, \mathbf{A}(j,:)^-]_{j=1}^p\}$ with shift $(\mathbf{B}(1,:)^+ + \mathbf{B}(2,:)^-)\mathbf{A}^-$.*

*Note that $\mathbf{A}^+ = \max(\mathbf{A}, 0)$ and $\mathbf{A}^- = \max(-\mathbf{A}, 0)$ where the $\max(.)$ is element-wise. The line segment $(\mathbf{B}(1,j)^+ + \mathbf{B}(2,j)^-)[\mathbf{A}(j,:)^+, \mathbf{A}(j,:)^-]$ is one that has the end points $\mathbf{A}(j,:)^+$ and $\mathbf{A}(j,:)^-$ in $\mathbb{R}^n$ and scaled by the constant $\mathbf{B}(1,j)^+ + \mathbf{B}(2,j)^-$.*

*Proof.* For the first part, recall from Theorem1 that both $f_1$ and $f_2$ are tropical rationals and hence,

$$f_1(\mathbf{x}) = H_1(\mathbf{x}) - Q_1(\mathbf{x}) \qquad f_2(\mathbf{x}) = H_2(\mathbf{x}) - Q_2(\mathbf{x})$$

Thus

$$\begin{aligned}
\mathcal{B} = \{x \in \mathbb{R}^n : f_1(\mathbf{x}) = f_2(\mathbf{x})\} &= \{x \in \mathbb{R}^n : H_1(\mathbf{x}) - Q_1(\mathbf{x}) = H_2(\mathbf{x}) - Q_2(\mathbf{x})\} \\
&= \{x \in \mathbb{R}^n : H_1(\mathbf{x}) + Q_2(\mathbf{x}) = H_2(\mathbf{x}) + Q_1(\mathbf{x})\} \\
&= \{x \in \mathbb{R}^n : H_1(\mathbf{x}) \odot Q_2(\mathbf{x}) = H_2(\mathbf{x}) \odot Q_1(\mathbf{x})\}
\end{aligned}$$

Recall that the tropical hypersurface is defined as the set of $\mathbf{x}$ where the maximum is attained by two or more monomials. Therefore, the tropical hypersurface of $R(\mathbf{x})$ is the set of $\mathbf{x}$ where the maximum is attained by two or more monomials in $(H_1(\mathbf{x}) \odot Q_2(\mathbf{x}))$, or attained by two or more monomials in $(H_2(\mathbf{x}) \odot Q_1(\mathbf{x}))$, or attained by monomials in both of them in the same time, which is the decision boundaries. Hence, we can rewrite that as

$$\mathcal{T}(R(\mathbf{x})) = \mathcal{T}(H_1(\mathbf{x}) \odot Q_2(\mathbf{x})) \cup \mathcal{T}(H_2(\mathbf{x}) \odot Q_1(\mathbf{x})) \cup \mathcal{B}.$$

Therefore $\mathcal{B} \subseteq \mathcal{T}(R(x))$. For the second part of the Theorem, we first use the decomposition proposed by Zhang et al. (2018); Berrada et al. (2016) to show that for a network $f(\mathbf{x}) = \mathbf{B} \max(\mathbf{Ax}, \mathbf{0})$, it can be decomposed as tropical rational as follows

$$\begin{aligned}
f(\mathbf{x}) &= (\mathbf{B}^+ - \mathbf{B}^-)\left(\max(\mathbf{A}^+\mathbf{x}, \mathbf{A}^-\mathbf{x}) - \mathbf{A}^-\mathbf{x}\right) \\
&= \left[\mathbf{B}^+ \max(\mathbf{A}^+\mathbf{x}, \mathbf{A}^-\mathbf{x}) + \mathbf{B}^-\mathbf{A}^-\mathbf{x}\right] - \left[\mathbf{B}^- \max(\mathbf{A}^+\mathbf{x}, \mathbf{A}^-\mathbf{x}) + \mathbf{B}^+\mathbf{A}^-\mathbf{x}\right].
\end{aligned}$$

Therefore, we have that

$$H_1(\mathbf{x}) + Q_2(\mathbf{x}) = \left(\mathbf{B}^+(1,:) + \mathbf{B}^-(2,:)\right)\max(\mathbf{A}^+\mathbf{x}, \mathbf{A}^-\mathbf{x}) + \left(\mathbf{B}^-(1,:) + \mathbf{B}^+(2,:)\right)\mathbf{A}^-\mathbf{x},$$

$$H_2(\mathbf{x}) + Q_1(\mathbf{x}) = \left(\mathbf{B}^-(1,:) + \mathbf{B}^+(2,:)\right)\max(\mathbf{A}^+\mathbf{x}, \mathbf{A}^-\mathbf{x}) + \left(\mathbf{B}^+(1,:) + \mathbf{B}^-(2,:)\right)\mathbf{A}^-\mathbf{x}.$$

Therefore note that

$$\begin{aligned}
\delta(R(\mathbf{x})) &= \delta\left(\left(H_1(\mathbf{x}) \odot Q_2(\mathbf{x})\right) \oplus \left(H_2(\mathbf{x}) \odot Q_1(\mathbf{x})\right)\right) \\
&\stackrel{(8)}{=} \text{ConvexHull}\left(\delta\left(H_1(\mathbf{x}) \odot Q_2(\mathbf{x})\right), \delta\left(H_2(\mathbf{x}) \odot Q_1(\mathbf{x})\right)\right) \\
&\stackrel{(7)}{=} \text{ConvexHull}\left(\delta\left(H_1(\mathbf{x})\right) \tilde{+} \delta\left(Q_2(\mathbf{x})\right), \delta\left(H_2(\mathbf{x})\right) \tilde{+} \delta\left(Q_1(\mathbf{x})\right)\right).
\end{aligned}$$

Now observe that $H_1(\mathbf{x}) = \sum_{j=1}^{p} \left( \mathbf{B}^+(1,j) + \mathbf{B}^-(2,j) \right) \max \left( \mathbf{A}^+(j,:), \mathbf{A}^-(j,:)\mathbf{x} \right)$ tropically is

given as follows $H_1(\mathbf{x}) = \odot_{j=1}^{p} \left[ \mathbf{x}^{\mathbf{A}^+(j,:)} \oplus \mathbf{x}^{\mathbf{A}^-(j,:)} \right]^{\mathbf{B}^+(1,j)\odot \mathbf{B}^-(2,j)}$ , thus we have that

$$\delta(H_1(\mathbf{x})) = \left( \mathbf{B}^+(1,1) + \mathbf{B}^-(2,1) \right) \delta \left( \mathbf{x}^{\mathbf{A}^+(1,:)} \oplus \mathbf{x}^{\mathbf{A}^-(1,:)} \right) \tilde{+} \dots$$
$$\tilde{+} \left( \mathbf{B}^+(1,p) + \mathbf{B}^-(2,p) \right) \left( \delta(\mathbf{x}^{\mathbf{A}^+(p,:)} \oplus \mathbf{x}^{\mathbf{A}^-(p,:)}) \right)$$
$$= \left( \mathbf{B}^+(1,1) + \mathbf{B}^-(2,1) \right) \text{ConvexHull} \left( \mathbf{A}^+(1,:), \mathbf{A}^-(1,:) \right) \tilde{+} \dots$$
$$\tilde{+} \left( \mathbf{B}^+(1,p) + \mathbf{B}^-(2,p) \right) \text{ConvexHull} \left( \mathbf{A}^+(p,:), \mathbf{A}^-(p,:) \right).$$

The operator $\tilde{+}$ indicates a Minkowski sum between sets. Note that $\text{ConvexHull} \left( \mathbf{A}^+(i,:), \mathbf{A}^-(i,:) \right)$ is the convexhull between two points which is a line segment in $\mathbb{Z}^n$ with end points that are $\{\mathbf{A}^+(i,:), \mathbf{A}^+(i,:)\}$ scaled with $\mathbf{B}^+(1,i) + \mathbf{B}^-(2,i)$. Observe that $\delta(F_1(\mathbf{x}))$ is a Minkowski sum of line segments which is is a zonotope. Moreover, note that $Q_2(\mathbf{x}) = (\mathbf{B}^-(1,:) + \mathbf{B}^+(2,:))\mathbf{A}^-\mathbf{x}$ tropically is given as follows $Q_2(\mathbf{x}) = \odot_{j=1}^{p} \mathbf{x}^{\mathbf{A}^-(j,:)(\mathbf{B}^+(1,j)\odot \mathbf{B}^-(2,j))}$. Thus it is easy to see that $\delta(Q_2(\mathbf{x}))$ is the Minkowski sum of the points $\{(\mathbf{B}^-(1,j) - \mathbf{B}^+(2,j))\mathbf{A}^-(j,:)\}\forall j$ in $\mathbb{R}^n$ (which is a standard sum) resulting in a point. Lastly, it is easy to see that $\delta(H_1(\mathbf{x}))\tilde{+}\delta(Q_2(\mathbf{x}))$ is a Minkowski sum between a zonotope and a single point which corresponds to a shifted zonotope. A similar symmetric argument can be applied for the second part $\delta(H_2(\mathbf{x}))\tilde{+}\delta(Q_1(\mathbf{x}))$.  □

It is also worthy to mention that the extension to network with multi class output is trivial. In that case all of the analysis can be exactly applied studying the decision boundary between any two classes $(i,j)$ where $\mathcal{B} = \{x \in \mathbb{R}^n : f_i(\mathbf{x}) = f_j(\mathbf{x})\}$ and the rest of the proof will be exactly the same.

## C    DERIVATION WITH BIASES

In this section, we derive the statement of Theorem 2 for the neural network in the form of (Affine, ReLU, Affine) with the consideration of non-zero biases. We show that the presence of biases does not affect the obtained results as they only increase the dimension of the space, where the polytopes live, without affecting their shape or edge-orientation. Starting with the first linear layer for $\mathbf{x} \in \mathbb{R}^n$, we have

$$\mathbf{z}_1 = \mathbf{A}\mathbf{x} + \mathbf{c}_1 = \mathbf{A}^+\mathbf{x} + \mathbf{c}_1 - \mathbf{A}^-\mathbf{x} = \mathbf{H}_1 \oslash \mathbf{Q}_1,$$

with coordinates

$$z_{1_i} = \mathbf{A}^+(i,:)\mathbf{x} + c_{1_i} - \mathbf{A}^-(i,:)\mathbf{x} = (c_{1_i} \odot \mathbf{x}^{\mathbf{A}^+(i,:)}) \oslash \mathbf{x}^{\mathbf{A}^-(i,:)} = \mathbf{H}_{1_i} \oslash \mathbf{Q}_{1_i}.$$

Thus, $\Delta(\mathbf{H}_{1_i})$ is a point in (n+1) dimensions at $(\mathbf{A}^+(i,:), c_{1_i})$, and $\Delta(\mathbf{Q}_{1_i})$ is a point in $(n+1)$ dimensions at $(\mathbf{A}^-(i,:), 0)$, while under $\pi$ projection, $\delta(\mathbf{H}_{1_i})$ is a point in $n$ dimensions at $(\mathbf{A}^+(i,:))$, and $\delta(\mathbf{Q}_{1_i})$ is a point in $n$ dimensions at $(\mathbf{A}^-(i,:))$. It can be seen that under projection $\pi$, the geometrical representation of the output of the first linear layer does not change after adding biases.

Looking to the output after adding the ReLU layer, we get

$$\mathbf{z}_2 = \max(\mathbf{z}_1, \mathbf{0}) = \max(\mathbf{A}^+\mathbf{x} + \mathbf{c}_1, \mathbf{A}^-\mathbf{x}) - \mathbf{A}^-\mathbf{x} = (\mathbf{H}_1 \oplus \mathbf{Q}_1) - \mathbf{Q}_1 = \mathbf{H}_2 \oslash \mathbf{Q}_2.$$

Hence, $\Delta(\mathbf{H}_{2_i})$ is the line segment $[(\mathbf{A}^+(i,:), c_{1_i}), (\mathbf{A}^-(i,:), \mathbf{0})]$, and $\Delta(\mathbf{Q}_{1_i})$ is the point $(\mathbf{A}^-(i,:), \mathbf{0})$. Thus, $\delta(\mathbf{H}_{2_i})$ is the line segment $[(\mathbf{A}^+(i,:)), (\mathbf{A}^-(i,:))]$, and $\delta(\mathbf{Q}_{1_i})$ is the point $(\mathbf{A}^-(i,:))$. Again, the biases does not affect the geometry of the output after the ReLU layer, since the line segments now are connecting points in $(n+1)$ dimensions, but after projecting them using $\pi$, they will be identical to the line segments of the network with zero biases.

Finally, looking to the output of the second linear layer, we obtain

$$\begin{aligned}
\mathbf{z}_3 &= \mathbf{B}\mathbf{z}_2 + \mathbf{c}_2 = (\mathbf{B}^+ - \mathbf{B}^-)(\mathbf{H}_2 - \mathbf{Q}_2) + \mathbf{c}_2 \\
&= (\mathbf{B}^+\mathbf{H}_2 + \mathbf{B}^-\mathbf{Q}_2 + \mathbf{c}_2) - (\mathbf{B}^-\mathbf{H}_2 + \mathbf{B}^+\mathbf{Q}_2) \\
&= \mathbf{H}_3 \oslash \mathbf{Q}_3
\end{aligned}$$

Therefore

$$\Delta(\mathbf{H}_{3_i}) = \tilde{+}_j(\Delta(\mathbf{B}(i,j)\mathbf{H}_2(j,:)))\tilde{+}\Delta\Big(\sum_j \mathbf{B}^-(i,j)\mathbf{Q}_2(j,:), c_{2_i}\Big)$$

$$\delta(\mathbf{H}_{3_i}) = \tilde{+}_j(\delta(\mathbf{B}(i,j)\mathbf{H}_2(j,:)))\tilde{+}\delta\Big(\sum_j \mathbf{B}^-(i,j)\mathbf{Q}_2(j,:)\Big)$$

Similar arguments can be given for $\Delta(\mathbf{Q}_{3_i})$ and $\delta(\mathbf{Q}_{3_i})$. It can be seen that the first part in both expressions is a Minkowski sum of line segments, which will give a zonotope in $(n+1)$, and $n$ dimensions in the first and second expressions respectively. While the second part in both expressions is a Minkowski sum of bunch of points which gives a single point in $(n+1)$ and $n$ dimensions for the first and second expression respectively. Note that the last dimension of the aforementioned point in $n+1$ dimensions is exactly the $i^{th}$ coordinate of the bias of the second linear layer which is dropped under the $\pi$ projection. Therefore, the shape of the geometrical representation of the decision boundaries with non-zero biases will not be affected under the projection $\pi$, and hence the presence of the biases will not affect any of the results of the paper.

## D    PROOF OF PROPOSITION 1

**Proposition 1.** *Consider $p$ line segments in $\mathbb{R}^n$ with two arbitrary end points as follows $\{[\mathbf{u}_1^i, \mathbf{u}_2^i]\}_{i=1}^p$. The zonotope formed by these line segments is equivalent to the zonotope formed be the line segments $\{[\mathbf{u}_1^i - \mathbf{u}_2^i, \mathbf{0}]\}_{i=1}^p$ with a shift of $\sum_{i=1}^p \mathbf{u}_2^i$.*

*Proof.* Let $\mathbf{U}_j$ be a matrix with $\mathbf{U}_j(:, i) = \mathbf{u}_j^i, i = 1, \ldots, p$, $\mathbf{w}$ be a column-vector with $\mathbf{w}(i) = w_i, i = 1, \ldots, p$ and $\mathbf{1}_p$ is a column-vector of ones of length $p$. Then, the zonotope $\mathcal{Z}$ formed by the Minkowski sum of line segments with arbitrary end points can be defined as

$$
\begin{aligned}
\mathcal{Z} &= \Big\{ \sum_{i=1}^p w_i \mathbf{u}_1^i + (1 - w_i)\mathbf{u}_2^i; w_i \in [0, 1], \ \forall\, i \Big\} \\
&= \Big\{ \mathbf{U}_1 \mathbf{w} - \mathbf{U}_2 \mathbf{w} + \mathbf{U}_2 \mathbf{1}_p, \ \ \mathbf{w} \in [0, 1]^p \Big\} \\
&= \Big\{ (\mathbf{U}_1 - \mathbf{U}_2)\, \mathbf{w} + \mathbf{U}_2 \mathbf{1}_p, \ \ \mathbf{w} \in [0, 1]^p \Big\} \\
&= \Big\{ (\mathbf{U}_1 - \mathbf{U}_2)\, \mathbf{w}, \ \ \mathbf{w} \in [0, 1]^p \Big\} \tilde{+} \Big\{ \mathbf{U}_2 \mathbf{1}_p \Big\}.
\end{aligned}
$$

Note that the Minkowski sum of any polytope with a point is a translation; thus, the result follows directly from Definition 6. □

### D.1 OPTIMIZATION OF OBJECTIVE EQUATION 2 OF THE BINARY CLASSIFIER

$$\min_{\tilde{\mathbf{A}}, \tilde{\mathbf{B}}} \frac{1}{2} \left\| \tilde{\mathbf{G}}_1 - \mathbf{G}_1 \right\|_F^2 + \left\| \frac{1}{2} \tilde{\mathbf{G}}_2 - \mathbf{G}_2 \right\|_F^2 + \lambda_1 \left\| \tilde{\mathbf{G}}_1 \right\|_{2,1} + \lambda_2 \left\| \tilde{\mathbf{G}}_2 \right\|_{2,1} . \tag{9}$$

Note that $\tilde{\mathbf{G}}_1 = \mathrm{Diag}\left[ \mathrm{ReLU}(\tilde{\mathbf{B}}(1,:)) + \mathrm{ReLU}(-\tilde{\mathbf{B}}(2,:)) \right] \tilde{\mathbf{A}}$, $\tilde{\mathbf{G}}_2 = \mathrm{Diag}\left[ \mathrm{ReLU}(\tilde{\mathbf{B}}(2,:)) + \mathrm{ReLU}(-\tilde{\mathbf{B}}(1,:)) \right] \tilde{\mathbf{A}}$. Note that $\mathbf{G}_1 = \mathrm{Diag}\left[ \mathrm{ReLU}(\mathbf{B}(1,:)) + \mathrm{ReLU}(-\mathbf{B}(2,:)) \right] \mathbf{A}$ and $\mathbf{G}_2 = \mathrm{Diag}\left[ \mathrm{ReLU}(\mathbf{B}(2,:)) + \mathrm{ReLU}(-\mathbf{B}(1,:)) \right] \mathbf{A}$. For ease of notation we refer to $\mathrm{ReLU}(\tilde{\mathbf{B}}(i,:))$ and $\mathrm{ReLU}(-\tilde{\mathbf{B}}(i,:))$ as $\tilde{\mathbf{B}}^+(i,:)$ and $\tilde{\mathbf{B}}^-(i,:)$, respectively. We solve the problem with co-rodinate descent an alternate over variables.

**Update $\tilde{\mathbf{A}}$.**

$$\tilde{\mathbf{A}} \leftarrow \arg\min_{\tilde{\mathbf{A}}} \frac{1}{2} \left\| \mathrm{Diag}\left( \mathbf{c}_1 \right) \tilde{\mathbf{A}} - \mathbf{G}_1 \right\|_F^2 + \frac{1}{2} \left\| \mathrm{Diag}(\mathbf{c}_2) \tilde{\mathbf{A}} - \mathbf{G}_2 \right\|_F^2 + \lambda_1 \left\| \mathrm{Diag}(\mathbf{c}_1) \tilde{\mathbf{A}} \right\|_{2,1} + \lambda_2 \left\| \mathrm{Diag}(\mathbf{c}_2) \tilde{\mathbf{A}} \right\|_{2,1},$$

where $\mathbf{c}_1 = \mathrm{ReLU}(\mathbf{B}(1,:)) + \mathrm{ReLU}(-\mathbf{B}(2,:))$ and $\mathbf{c}_2 = \mathrm{ReLU}(\mathbf{B}(2,:)) + \mathrm{ReLU}(-\mathbf{B}(1,:))$. Note that the problem is separable per-row of $\tilde{\mathbf{A}}$. Therefore, the problem reduces to updating rows of $\tilde{\mathbf{A}}$ independently and the problem exhibits a closed form solution.

$$
\begin{aligned}
\tilde{\mathbf{A}}(i,:) &= \arg\min_{\tilde{\mathbf{A}}(i,:)} \frac{1}{2} \left\| \mathbf{c}_1^i \tilde{\mathbf{A}}(i,:) - \mathbf{G}_1(i,:) \right\|_2^2 + \frac{1}{2} \left\| \mathbf{c}_2^i \tilde{\mathbf{A}}(i,:) - \mathbf{G}_2(i,:) \right\|_2^2 + (\lambda_1 \sqrt{\mathbf{c}_1^i} + \lambda_2 \sqrt{\mathbf{c}_2^i}) \left\| \tilde{\mathbf{A}}(i,:) \right\|_2 \\
&= \arg\min_{\tilde{\mathbf{A}}(i,:)} \frac{1}{2} \left\| \tilde{\mathbf{A}}(i,:) - \frac{\mathbf{c}_1^i \mathbf{G}_1(i,:) + \mathbf{c}_2^i \mathbf{G}_2(i,:)}{\frac{1}{2}(\mathbf{c}_1^i + \mathbf{c}_2^i)} \right\|_2^2 + \frac{1}{2} \frac{\lambda_1 \sqrt{\mathbf{c}_1^i} + \lambda_2 \sqrt{\mathbf{c}_2^i}}{\frac{1}{2}(\mathbf{c}_1^i + \mathbf{c}_2^i)} \left\| \tilde{\mathbf{A}}(i,:) \right\|_2 \\
&= \left( 1 - \frac{1}{2} \frac{\lambda_1 \sqrt{\mathbf{c}_1^i} + \lambda_2 \sqrt{\mathbf{c}_2^i}}{\frac{1}{2}(\mathbf{c}_1^i + \mathbf{c}_2^i)} \frac{1}{\left\| \frac{\mathbf{c}_1^i \mathbf{G}_1(i,:) + \mathbf{c}_2^i \mathbf{G}_2(i,:)}{\frac{1}{2}(\mathbf{c}_1^i + \mathbf{c}_2^i)} \right\|_2} \right) \left( \frac{\mathbf{c}_1^i \mathbf{G}_1(i,:) + \mathbf{c}_2^i \mathbf{G}_2(i,:)}{\frac{1}{2}(\mathbf{c}_1^i + \mathbf{c}_2^i)} \right).
\end{aligned}
$$

**Update $\tilde{\mathbf{B}}^+(1,:)$.**

$$\tilde{\mathbf{B}}^+(1,:) = \arg\min_{\tilde{\mathbf{B}}^+(1,:)} \frac{1}{2} \left\| \mathrm{Diag}\left( \tilde{\mathbf{B}}^+(1,:) \right) \tilde{\mathbf{A}} - \mathbf{C}_1 \right\|_F^2 + \lambda_1 \left\| \mathrm{Diag}\left( \tilde{\mathbf{B}}^+(1,:) \right) \tilde{\mathbf{A}} + \mathbf{C}_2 \right\|_{2,1}, \quad \text{s.t. } \tilde{\mathbf{B}}^+(1,:) \geq \mathbf{0}.$$

Note that $\mathbf{C}_1 = \mathbf{G}_1 - \mathrm{Diag}\left( \tilde{\mathbf{B}}^-(2,:) \right) \tilde{\mathbf{A}}$ and where $\mathrm{Diag}\left( \tilde{\mathbf{B}}^-(2,:) \right) \tilde{\mathbf{A}}$. Note the problem is separable in the coordinates of $\tilde{\mathbf{B}}^+(1,:)$ and a projected gradient descent can be used to solve the problem in such a way as

$$\tilde{\mathbf{B}}^+(1,j) = \arg\min_{\tilde{\mathbf{B}}^+(1,j)} \frac{1}{2} \left\| \tilde{\mathbf{B}}^+(1,j)\tilde{\mathbf{A}}(j,:) - \mathbf{C}_1(j,:) \right\|_2^2 + \lambda_1 \left\| \tilde{\mathbf{B}}^+(1,j)\tilde{\mathbf{A}}(j,:) + \mathbf{C}_2(j,:) \right\|_2, \quad \text{s.t. } \tilde{\mathbf{B}}^+(1,j) \geq 0.$$

A similar symmetric argument can be used to update the variables $\tilde{\mathbf{B}}^+(2,:)$, $\tilde{\mathbf{B}}^+(1,:)$ and $\tilde{\mathbf{B}}^-(2,:)$.

## D.2 ADAPTING OPTIMIZATION EQUATION 2 FOR MULTI-CLASS CLASSIFIER

Note that Theorem 2 describes a superset to the decision boundaries of a binary classifier through the dual subdivision $R(\mathbf{x})$, *i.e.* $\delta(R(\mathbf{x}))$. For a neural network $f$ with $k$ classes, a natural extension for it is to analyze the pair-wise decision boundaries of of all $k$-classes. Thus, let $\mathcal{T}(R_{ij}(\mathbf{x}))$ be the superset to the decision boundaries separating classes $i$ and $j$. Therefore, a natural extension to the geometric loss in equation 1 is to preserve the polytopes among all pairwise follows

$$\min_{\tilde{\mathbf{A}},\tilde{\mathbf{B}}} \sum_{\forall [i,j] \in S} d\Big(\text{ConvexHull}\left(\mathcal{Z}_{\tilde{\mathbf{G}}_{(i^+,j^-)}}, \mathcal{Z}_{\tilde{\mathbf{G}}_{(j^+,i^-)}}\right), \text{ConvexHull}\left(\mathcal{Z}_{\mathbf{G}_{(i^+,j^-)}}, \mathcal{Z}_{\mathbf{G}_{(j^+,i^-)}}\right)\Big).$$
(10)

The set $S$ is all possible pairwise combinations of the $k$ classes such that $S = \{[i,j], \forall i \neq j, i = 1, \ldots, k, j = 1, \ldots, k\}$. The generator $\mathcal{Z}(\tilde{G}_{(i,j)})$ is the zonotope with the generator matrix $\tilde{\mathbf{G}}_{(i^+,j^-)} = \text{Diag}\left[\text{ReLU}(\tilde{\mathbf{B}}(i,:)) + \text{ReLU}(-\tilde{\mathbf{B}}(j,:))\right] \tilde{\mathbf{A}}$. However, such an approach is generally computationally expensive, particularly, when $k$ is very large. To this end, we make the following observation that $\tilde{\mathbf{G}}_(i^+, j^-)$ can be equivalently written as a Minkowski sum between two sets zonotopes with the generators $\mathbf{G}_{(i^+)} = \text{Diag}\left[\text{ReLU}(\tilde{\mathbf{B}}(i,:)\right] \tilde{\mathbf{A}}$ and $\mathbf{G}_{j^-} = \text{Diag}\left[\text{ReLU}(\tilde{\mathbf{B}}_{j^-})\right] \tilde{\mathbf{A}}$. That is to say, $\mathcal{Z}_{\tilde{\mathbf{G}}_{(i^+,j^-)}} = \mathcal{Z}_{\tilde{\mathbf{G}}_{i^+}} \tilde{+} \mathcal{Z}_{\tilde{\mathbf{G}}_{j^-}}$. This follows from the associative property of Minkowski sums given as follows:

**Fact 5.** *Let* $\{S_i\}_{i=1}^n$ *be the set of* $n$ *line segments. Then we have that*

$$S = S_1 \tilde{+} \ldots \tilde{+} S_n = P \tilde{+} V$$

*where the sets* $P = \tilde{+}_{j \in C_1} S_j$ *and* $V = \tilde{+}_{j \in C_2} S_j$ *where* $C_1$ *and* $C_2$ *are any complementary partitions of the set* $\{S_i\}_{i=1}^n$.

Hence, $\tilde{\mathbf{G}}_{(i^+,j^-)}$ can be seen a concatenation between $\tilde{\mathbf{G}}_(i^+)$ and $\tilde{\mathbf{G}}_(j^-)$. Thus, the objective in 10 can be expanded as follows

$$\min_{\tilde{\mathbf{A}},\tilde{\mathbf{B}}} \sum_{\forall [i,j] \in S} d\Big(\text{ConvexHull}\left(\mathcal{Z}_{\tilde{\mathbf{G}}_{(i^+,j^-)}}, \mathcal{Z}_{\tilde{\mathbf{G}}_{(j^+,i^-)}}\right), \text{ConvexHull}\left(\mathcal{Z}_{\mathbf{G}_{(i^+,j^-)}}, \mathcal{Z}_{\mathbf{G}_{(j^+,i^-)}}\right)\Big)$$

$$= \min_{\tilde{\mathbf{A}},\tilde{\mathbf{B}}} \sum_{\forall [i,j] \in S} d\Big(\text{ConvexHull}\left(\mathcal{Z}_{\tilde{\mathbf{G}}_{i^+}} \tilde{+} \mathcal{Z}_{\tilde{\mathbf{G}}_{j^-}}, \mathcal{Z}_{\tilde{\mathbf{G}}_j^+} \tilde{+} \mathcal{Z}_{\tilde{\mathbf{G}}_{i^-}}\right), \text{ConvexHull}\left(\mathcal{Z}_{\mathbf{G}_{i^+}} \tilde{+} \mathcal{Z}_{\mathbf{G}_{j^-}}, \mathcal{Z}_{\mathbf{G}_j^+} \tilde{+} \mathcal{Z}_{\mathbf{G}_{i^-}}\right)\Big)$$

$$\approx \min_{\tilde{\mathbf{A}},\tilde{\mathbf{B}}} \sum_{\forall [i,j] \in S} \left\| \begin{pmatrix} \tilde{\mathbf{G}}_{i^+} \\ \mathbf{G}_{j^-} \end{pmatrix} - \begin{pmatrix} \tilde{\mathbf{G}}_{i^+} \\ \mathbf{G}_{j^-} \end{pmatrix} \right\|_F^2 + \left\| \begin{pmatrix} \tilde{\mathbf{G}}_{i^-} \\ \mathbf{G}_{j^+} \end{pmatrix} - \begin{pmatrix} \tilde{\mathbf{G}}_{i^-} \\ \mathbf{G}_{j^+} \end{pmatrix} \right\|_F^2$$

$$= \min_{\tilde{\mathbf{A}},\tilde{\mathbf{B}}} \sum_{\forall [i,j] \in S} \frac{1}{2}\left\|\tilde{\mathbf{G}}_{i^+} - \mathbf{G}_{i^+}\right\|_F^2 + \frac{1}{2}\left\|\tilde{\mathbf{G}}_{i^-} - \mathbf{G}_{i^-}\right\|_F^2 + \frac{1}{2}\left\|\tilde{\mathbf{G}}_{j^+} - \mathbf{G}_{j^+}\right\|_F^2 + \frac{1}{2}\left\|\tilde{\mathbf{G}}_{j^-} - \mathbf{G}_{j^-}\right\|_F^2$$

$$= \min_{\tilde{\mathbf{A}},\tilde{\mathbf{B}}} \sum_{i=1}^{k} \frac{1}{2}(k-1)\left(\left\|\tilde{\mathbf{G}}_{i^+} - \mathbf{G}_{i^+}\right\|_F^2 + \left\|\tilde{\mathbf{G}}_{i^-} - \mathbf{G}_{i^-}\right\|_F^2 + \left\|\tilde{\mathbf{G}}_{j^+} - \mathbf{G}_{j^+}\right\|_F^2 + \left\|\tilde{\mathbf{G}}_{j^-} - \mathbf{G}_{j^-}\right\|_F^2\right).$$

The approximation follows in a similar argument to the binary classifier case where approximating the generators. The last equality follows from a counting argument. We solve the objective for all multi-class networks in the experiments with alternating optimization in a similar fashion to the binary classifier case. Similarly to the binary classification approach, we introduce the $\|\|_{2,1}$ to enforce sparsity constraints for pruning purposes. Therefore the overall objective has the form

$$\min_{\tilde{\mathbf{A}},\tilde{\mathbf{B}}} \sum_{i=1}^{k} \frac{1}{2}\left(\left\|\tilde{\mathbf{G}}_{i^+} - \mathbf{G}_{i^+}\right\|_F^2 + \left\|\tilde{\mathbf{G}}_{i^-} - \mathbf{G}_{i^-}\right\|_F^2 + \left\|\tilde{\mathbf{G}}_{j^+} - \mathbf{G}_{j^+}\right\|_F^2 + \left\|\tilde{\mathbf{G}}_{j^-} - \mathbf{G}_{j^-}\right\|_F^2\right)$$

$$+ \lambda\left(\left\|\tilde{\mathbf{G}}_{i^+}\right\|_{2,1} + \left\|\tilde{\mathbf{G}}_{i^-}\right\|_{2,1} + \left\|\tilde{\mathbf{G}}_{j^+}\right\|_{2,1} + \left\|\tilde{\mathbf{G}}_{j^-}\right\|_{2,1}\right).$$

For completion, we derive the updates for $\tilde{\mathbf{A}}$ and $\tilde{\mathbf{B}}$.

**Update $\tilde{\mathbf{A}}$.**

$$\tilde{\mathbf{A}} = \arg\min_{\tilde{\mathbf{A}}} \sum_{i=1}^{k} \frac{1}{2} \left( \left\| \text{Diag}\left( \tilde{\mathbf{B}}^+(i,:) \right) \tilde{\mathbf{A}} - \mathbf{G}_{i+} \right\|_F^2 + \left\| \text{Diag}\left( \tilde{\mathbf{B}}^-(i,:) \right) \tilde{\mathbf{A}} - \mathbf{G}_{i-} \right\|_F^2 \right.$$

$$+ \left\| \text{Diag}\left( \tilde{\mathbf{B}}^+(j,:) \right) \tilde{\mathbf{A}} - \mathbf{G}_{j+} \right\|_F^2 + \left\| \text{Diag}\left( \tilde{\mathbf{B}}^-(j,:) \right) \tilde{\mathbf{A}} - \mathbf{G}_{j-} \right\|_F^2 \right)$$

$$+ \lambda \left( \left\| \text{Diag}\left( \tilde{\mathbf{B}}^+(i,:) \right) \tilde{\mathbf{A}} \right\|_{2,1} + \left\| \text{Diag}\left( \tilde{\mathbf{B}}^-(i,:) \right) \tilde{\mathbf{A}} \right\|_{2,1} + \left\| \text{Diag}\left( \tilde{\mathbf{B}}^+(j,:) \right) \tilde{\mathbf{A}} \right\|_{2,1} \right.$$

$$+ \left\| \text{Diag}\left( \tilde{\mathbf{B}}^-(j,:) \right) \tilde{\mathbf{A}} \right\|_{2,1} \right).$$

Similar to the binary classification, the problem is seprable in the rows of $\tilde{\mathbf{A}}$. and a closed form solution in terms of the proximal operator of $\ell_2$ norm follows naturally for each $\tilde{\mathbf{A}}(i,:)$.

**Update $\tilde{\mathbf{B}}^+(i,:)$.**

$$\tilde{\mathbf{B}}^+(i,:) = \arg\min_{\tilde{\mathbf{B}}^+(i,:)} \frac{1}{2} \left\| \text{Diag}\left( \tilde{\mathbf{B}}^+(i,:) \right) \tilde{\mathbf{A}} - \tilde{\mathbf{G}}_{i+} \right\|_F^2 + \lambda \left\| \text{Diag}\left( \tilde{\mathbf{B}}^+(i,:) \right) \tilde{\mathbf{A}} \right\|_{2,1}, \quad \text{s.t. } \tilde{\mathbf{B}}^+(i,:) \geq \mathbf{0}.$$

Note that the problem is separable per coordinates of $\mathbf{B}^+(i,:)$ and each subproblem is updated as:

$$\tilde{\mathbf{B}}^+(i,j) = \arg\min_{\tilde{\mathbf{B}}^+(i,j)} \frac{1}{2} \left\| \tilde{\mathbf{B}}^+(i,j) \tilde{\mathbf{A}}(j,:) - \tilde{\mathbf{G}}_{i+}(j,:) \right\|_2^2 + \lambda \left\| \tilde{\mathbf{B}}^+(i,j) \tilde{\mathbf{A}}(j,:) \right\|_2, \quad \text{s.t. } \tilde{\mathbf{B}}^+(i,j) \geq 0$$

$$= \arg\min_{\tilde{\mathbf{B}}^+(i,j)} \frac{1}{2} \left\| \tilde{\mathbf{B}}^+(i,j) \tilde{\mathbf{A}}(j,:) - \tilde{\mathbf{G}}_{i+}(j,:) \right\|_2^2 + \lambda \left| \tilde{\mathbf{B}}(i,j) \right| \left\| \tilde{\mathbf{A}}(j,:) \right\|_2, \quad \text{s.t. } \tilde{\mathbf{B}}^+(i,j) \geq 0$$

$$= \max\left( 0, \frac{\tilde{\mathbf{A}}(j,:)^\top \tilde{\mathbf{G}}_{i+}(j,:) - \lambda \|\tilde{\mathbf{A}}(j,:)\|_2}{\|\tilde{\mathbf{A}}(j,:)\|_2^2} \right).$$

A similar argument can be used to update $\tilde{\mathbf{B}}^-(i,:) \ \forall i$. Finally, the parameters of the pruned network will be constructed $\mathbf{A} \leftarrow \tilde{\mathbf{A}}$ and $\mathbf{B} \leftarrow \tilde{\mathbf{B}}^+ - \tilde{\mathbf{B}}^-$.

---

**Algorithm 1:** Solving Problem (5)

---

**Input** : $\mathbf{A}_1 \in \mathbb{R}^{p \times n}, \mathbf{B} \in \mathbb{R}^{k \times p}, \mathbf{x}_0 \in \mathbb{R}^n, t, \lambda > 0, \gamma > 1, K > 0, \xi_{\mathbf{A}_1} = \mathbf{0}_{p \times n}, \eta^1 = \mathbf{z}^1 = \mathbf{w}^1 = \mathbf{z}^1 = \mathbf{u}^1 = \mathbf{w}^1 = \mathbf{0}_n.$

**Output:** $\eta, \xi_{\mathbf{A}_1}$

**Initialize:** $\rho = \rho_0$

  **while** not converged **do**

    **for** k $\leq$ K **do**

      $\eta$ **update:** $\eta^{k+1} = (2\lambda \mathbf{A}_1^\top \mathbf{A}_1 + (2 + \rho)\mathbf{I})^{-1}(2\lambda \mathbf{A}_1^\top \xi_{\mathbf{A}_1}^k \mathbf{x}_0 + \rho \mathbf{z}^k - \mathbf{u}^k)$

      **w update:** $\mathbf{w}^{k+1} = \begin{cases} \min(1 - \mathbf{x}_0, \epsilon_1) & : \mathbf{z}^k - {}^1/\rho \mathbf{v}^k > \min(1 - \mathbf{x}_0, \epsilon_1) \\ \max(-\mathbf{x}_0, -\epsilon_1) & : \mathbf{z}^k - {}^1/\rho \mathbf{v}^k < \max(-\mathbf{x}_0, -\epsilon_1) \\ \mathbf{z}^k - {}^1/\rho \mathbf{v}^k & : otherwise \end{cases}$

      **z update:** $\mathbf{z}^{k+1} = \frac{1}{\eta^{k+1}+2\rho}(\eta^{k+1}\mathbf{z}^k + \rho(\eta^{k+1} + {}^1/\rho \mathbf{u}^k + \mathbf{w}^k + {}^1/\rho \mathbf{v}^k) - \nabla \mathcal{L}(\mathbf{z}^k + \mathbf{x}_0))$

      $\xi_{\mathbf{A}_1}$ **update:**

      $\xi_{\mathbf{A}_1}^{k+1} = \arg\min_{\xi_{\mathbf{A}}} \|\xi_{\mathbf{A}_1}\|_F^2 + \lambda \|\xi_{\mathbf{A}_1}\mathbf{x}_0 - \mathbf{A}_1 \eta^{k+1}\|_2^2 + \bar{\mathcal{L}}(\mathbf{A}_1)$ s.t. $\|\xi_{\mathbf{A}_1}\|_{\infty, \infty} \leq \epsilon_2$

      **u update:** $\mathbf{u}^{k+1} = \mathbf{u}^k + \rho(\eta^{k+1} - \mathbf{z}^{k+1})$

      **v update:** $\mathbf{v}^{k+1} = \mathbf{v}^k + \rho(\mathbf{w}^{k+1} - \mathbf{z}^{k+1}))$

      $\rho \leftarrow \gamma \rho$

    **end**

    $\lambda \leftarrow \gamma \lambda$

    $\rho \leftarrow \rho_0$

**end**

---

## E ALGORITHM FOR SOLVING 5.

In this section, we are going to derive an algorithm for solving the following problem.

$$\min_{\eta, \xi_{\mathbf{A}_1}} \quad \mathcal{D}_1(\eta) + \mathcal{D}_2(\xi_{\mathbf{A}_1})$$

$$\text{s.t.} \quad -loss(g(\mathbf{A}_1(\mathbf{x}_0 + \eta)), t) \leq -1, \quad -loss(g(\mathbf{A}_1 + \xi_{\mathbf{A}_1})\mathbf{x}_0, t) \leq -1, \tag{11}$$

$$(\mathbf{x}_0 + \eta) \in [0, 1]^n, \quad \|\eta\|_\infty \leq \epsilon_1, \quad \|\xi_{\mathbf{A}_1}\|_{\infty, \infty} \leq \epsilon_2, \quad \mathbf{A}_1 \eta - \xi_{\mathbf{A}_1}\mathbf{x}_0 = 0.$$

The function $\mathcal{D}_2(\xi_{\mathbf{A}})$ captures the perturbdation in the dual subdivision polytope such that the dual subdivion of the network with the first linear layer $\mathbf{A}_1$ is similar to the dual subdivion of the network with the first linear layer $\mathbf{A}_1 + \xi_{\mathbf{A}_1}$. This can be generally formulated as an approximation to the following distance function $d\left(\text{ConvHull}\left(\mathcal{Z}_{\tilde{\mathbf{G}}_1}, \mathcal{Z}_{\tilde{\mathbf{G}}_2}\right), \text{ConvHull}\left(\mathcal{Z}_{\mathbf{G}_1}, \mathcal{Z}_{\mathbf{G}_2}\right)\right)$, where $\tilde{\mathbf{G}}_1 = \text{Diag}\left[\text{ReLU}(\tilde{\mathbf{B}}(1, :)) + \text{ReLU}(-\tilde{\mathbf{B}}(2, :))\right]\left(\tilde{\mathbf{A}} + \xi_{\mathbf{A}_1}\right)$, $\tilde{\mathbf{G}}_2 = \text{Diag}\left[\text{ReLU}(\tilde{\mathbf{B}}(2, :)) + \text{ReLU}(-\tilde{\mathbf{B}}(1, :))\right]\left(\tilde{\mathbf{A}} + \xi_{\mathbf{A}_1}\right)$, $\mathbf{G}_1 = \text{Diag}\left[\text{ReLU}(\tilde{\mathbf{B}}(1, :)) + \text{ReLU}(-\tilde{\mathbf{B}}(2, :))\right]\tilde{\mathbf{A}}$ and $\mathbf{G}_2 = \text{Diag}\left[\text{ReLU}(\tilde{\mathbf{B}}(2, :)) + \text{ReLU}(-\tilde{\mathbf{B}}(1, :))\right]\tilde{\mathbf{A}}$. In particular, to approximate the function $d$, one can use a similar argument as in used in network pruning 5 such that $\mathcal{D}_2$ approximates the generators of the zonotopes directly as follows

$$\mathcal{D}_2(\xi_{\mathbf{A}_1}) = \frac{1}{2}\left\|\tilde{\mathbf{G}}_1 - \mathbf{G}_1\right\|_F^2 + \frac{1}{2}\left\|\tilde{\mathbf{G}}_2 - \mathbf{G}_2\right\|_F^2$$

$$= \frac{1}{2}\left\|\text{Diag}\left(\mathbf{B}^+(1, :)\right)\xi_{\mathbf{A}_1}\right\|_F^2 + \frac{1}{2}\left\|\text{Diag}\left(\mathbf{B}^-(1, :)\right)\xi_{\mathbf{A}_1}\right\|_F^2$$

$$+ \frac{1}{2}\left\|\text{Diag}\left(\mathbf{B}^+(2, :)\right)\xi_{\mathbf{A}_1}\right\|_F^2 + \frac{1}{2}\left\|\text{Diag}\left(\mathbf{B}^-(2, :)\right)\xi_{\mathbf{A}_1}\right\|_F^2.$$

This can thereafter be extended to multi-class network with $k$ classes as follows $\mathcal{D}_2(\xi_{\mathbf{A}_1}) = \frac{1}{2}\sum_{j=1}^k \left\|\text{Diag}\left(\mathbf{B}^+(j, :)\right)\xi_{\mathbf{A}_1}\right\|_F^2 + \left\|\text{Diag}\left(\mathbf{B}^-(j, :)\right)\xi_{\mathbf{A}_1}\right\|_F^2$. Following Xu et al. (2018), we take $\mathcal{D}_1(\eta) = \frac{1}{2}\|\eta\|_2^2$. Therefore, we can write 11 as follows

$$\min_{\eta, \xi_{\mathbf{A}}} \quad \mathcal{D}_1(\eta) + \sum_{j=1}^{k} \left\| \mathrm{Diag}\left(\mathbf{B}^+(j,:)\right) \xi_{\mathbf{A}} \right\|_F^2 + \left\| \mathrm{Diag}\left(\mathbf{B}^-(j,:)\right) \xi_{\mathbf{A}} \right\|_F^2.$$

$$\text{s.t.} \quad -loss(g(\mathbf{A}_1(\mathbf{x}_0 + \eta)), t) \leq -1, \quad -loss(g((\mathbf{A}_1 + \xi_{\mathbf{A}_1})\mathbf{x}_0), t) \leq -1,$$
$$(\mathbf{x}_0 + \eta) \in [0,1]^n, \quad \|\eta\|_\infty \leq \epsilon_1, \quad \|\xi_{\mathbf{A}_1}\|_{\infty,\infty} \leq \epsilon_2, \quad \mathbf{A}_1 \eta - \xi_{\mathbf{A}_1} \mathbf{x}_0 = 0.$$

To enforce the linear equality constraints $\mathbf{A}_1 \eta - \xi_{\mathbf{A}_1} \mathbf{x}_0 = 0$, we use a penalty method, where each iteration of the penalty method we solve the sub-problem with ADMM updates. That is, we solve the following optimization problem with ADMM with increasing $\lambda$ such that $\lambda \to \infty$. For ease of notation, lets denote $\mathcal{L}(\mathbf{x}_0 + \eta) = -loss(g(\mathbf{A}_1(\mathbf{x}_0 + \eta)), t)$, and $\bar{\mathcal{L}}(\mathbf{A}_1) = -loss(g((\mathbf{A}_1 + \xi_{\mathbf{A}_1})\mathbf{x}_0), t)$.

$$\min_{\eta, \mathbf{z}, \mathbf{w}, \xi_{\mathbf{A}_1}} \quad \|\eta\|_2^2 + \sum_{j=1}^{k} \left\| \mathrm{Diag}\left(\mathrm{ReLU}(\mathbf{B}(j,:))\right) \xi_{\mathbf{A}_1} \right\|_F^2 + \left\| \mathrm{Diag}\left(\mathrm{ReLU}(-\mathbf{B}(j,:))\right) \xi_{\mathbf{A}_1} \right\|_F^2$$

$$+ \mathcal{L}(\mathbf{x}_0 + \mathbf{z}) + h_1(\mathbf{w}) + h_2(\xi_{\mathbf{A}_1}) + \lambda \|\mathbf{A}_1 \eta - \xi_{\mathbf{A}_1} \mathbf{x}_0\|_2^2 + \bar{\mathcal{L}}(\mathbf{A}_1).$$
$$\text{s.t.} \quad \eta = \mathbf{z} \quad \mathbf{z} = \mathbf{w}.$$

where

$$h_1(\eta) = \begin{cases} 0, & \text{if } (\mathbf{x}_0 + \eta) \in [0,1]^n, \|\eta\|_\infty \leq \epsilon_1 \\ \infty, & \text{else} \end{cases} \qquad h_2(\xi_{\mathbf{A}_1}) = \begin{cases} 0, & \text{if } \|\xi_{\mathbf{A}_1}\|_{\infty,\infty} \leq \epsilon_2 \\ \infty, & \text{else} \end{cases}.$$

The augmented Lagrangian is thus given as follows

$$\mathcal{L}(\eta, \mathbf{w}, \mathbf{z}, \xi_{\mathbf{A}_1}, \mathbf{u}, \mathbf{v}) := \quad \|\eta\|_2^2 + \mathcal{L}(\mathbf{x}_0 + \mathbf{z}) + h_1(\mathbf{w}) + \sum_{j=1}^{k} \left\| \mathrm{Diag}(\mathbf{B}^+(j,:)) \xi_{\mathbf{A}_1} \right\|_F^2 + \left\| \mathrm{Diag}(\mathbf{B}^-(j,:)) \xi_{\mathbf{A}_1} \right\|_F^2$$

$$+ \bar{\mathcal{L}}(\mathbf{A}_1) + h_2(\xi_{\mathbf{A}_1}) + \lambda \|\mathbf{A}_1 \eta - \xi_{\mathbf{A}_1} \mathbf{x}_0\|_2^2 + \mathbf{u}^\top(\eta - \mathbf{z}) + \mathbf{v}^\top(\mathbf{w} - \mathbf{z})$$

$$+ \frac{\rho}{2}(\|\eta - \mathbf{z}\|_2^2 + \|\mathbf{w} - \mathbf{z}\|_2^2).$$

Thereafter, ADMM updates are given as follows

$$\{\eta^{k+1}, \mathbf{w}^{k+1}\} = \arg\min_{\eta, \mathbf{w}} \mathcal{L}(\eta, \mathbf{w}, \mathbf{z}^k, \xi_{\mathbf{A}_1}^k, \mathbf{u}^k, \mathbf{v}^k),$$

$$\mathbf{z}^{k+1} = \arg\min_{\mathbf{z}} \mathcal{L}(\eta^{k+1}, \mathbf{w}^{k+1}, \mathbf{z}, \xi_{\mathbf{A}_1}^k, \mathbf{u}^k, \mathbf{v}^k),$$

$$\xi_{\mathbf{A}_1}^{k+1} = \arg\min_{\xi_{\mathbf{A}_1}} \mathcal{L}(\eta^{k+1}, \mathbf{w}^{k+1}, \mathbf{z}^{k+1}, \xi_{\mathbf{A}_1}, \mathbf{u}^k, \mathbf{v}^k).$$

$$\mathbf{u}^{k+1} = \mathbf{u}^k + \rho(\eta^{k+1} - \mathbf{z}^{k+1}), \quad \mathbf{v}^{k+1} = \mathbf{v}^k + \rho(\mathbf{w}^{k+1} - \mathbf{z}^{k+1}).$$

**Updating $\eta$:**

$$\eta^{k+1} = \arg\min_{\eta} \|\eta\|_2^2 + \lambda \|\mathbf{A}_1 \eta - \xi_{\mathbf{A}_1} \mathbf{x}_0\|_2^2 + \mathbf{u}^\top \eta + \frac{\rho}{2} \|\eta - \mathbf{z}\|_2^2$$

$$= \left(2\lambda \mathbf{A}_1^\top \mathbf{A}_1 + (2 + \rho)\mathbf{I}\right)^{-1} \left(2\lambda \mathbf{A}_1^\top \xi_{\mathbf{A}_1}^k \mathbf{x}_0 + \rho \mathbf{z}^k - \mathbf{u}^k\right).$$

**Updating w:**

$$\mathbf{w}^{k+1} = \arg\min_{\mathbf{w}} {\mathbf{v}^k}^\top \mathbf{w} + h_1(\mathbf{w}) + \frac{\rho}{2}\|\mathbf{w} - \mathbf{z}^k\|_2^2$$

$$= \arg\min_{\mathbf{w}} \frac{1}{2}\left\|\mathbf{w} - \left(\mathbf{z}^k - \frac{\mathbf{v}^k}{\rho}\right)\right\|_2^2 + \frac{1}{\rho}h_1(\mathbf{w}).$$

It is easy to show that the update $\mathbf{w}$ is separable in coordinates as follows

$$\mathbf{w}^{k+1} = \begin{cases} \min(1 - \mathbf{x}_0, \epsilon_1) & : \mathbf{z}^k - \frac{1}{\rho}\mathbf{v}^k > \min(1 - \mathbf{x}_0, \epsilon_1) \\ \max(-\mathbf{x}_0, -\epsilon_1) & : \mathbf{z}^k - \frac{1}{\rho}\mathbf{v}^k < \max(-\mathbf{x}_0, -\epsilon_1) \\ \mathbf{z}^k - \frac{1}{\rho}\mathbf{v}^k & : \textit{otherwise} \end{cases}$$

**Updating $\mathbf{z}$:**

$$\mathbf{z}^{k+1} = \arg\min_{\mathbf{z}} \mathcal{L}(\mathbf{x}_0 + \mathbf{z}) - {\mathbf{u}^k}^\top \mathbf{z} - {\mathbf{v}^k}^\top \mathbf{z} + \frac{\rho}{2}\left(\|\eta^{k+1} - \mathbf{z}\|_2^2 + \|\mathbf{w}^{k+1} - \mathbf{z}\|_2^2\right).$$

Liu et al. (2019) showed that the linearized ADMM converges for some non-convex problems. Therefore, by linearizing $\mathcal{L}$ and adding Bergman divergence term $\eta^k/2\|\mathbf{z} - \mathbf{z}^k\|_2^2$, we can then update $z$ as follows

$$\mathbf{z}^{k+1} = \frac{1}{\eta^k + 2\rho}\left(\eta^k \mathbf{z}^k + \rho\left(\eta^{k+1} + \frac{1}{\rho}\mathbf{u}^k + \mathbf{w}^{k+1} + \frac{1}{\rho}\mathbf{v}^k\right) - \nabla\mathcal{L}(\mathbf{z}^k + \mathbf{x}_0)\right).$$

It is worthy to mention that the analysis until this step is inspired by Xu et al. (2018) with modifications to adapt our new formulation.

**Updating $\xi_{\mathbf{A}}$:**

$$\xi_{\mathbf{A}}^{k+1} = \arg\min_{\xi_{\mathbf{A}}} \|\xi_{\mathbf{A}_1}\|_F^2 + \lambda\|\xi_{\mathbf{A}_1}\mathbf{x}_0 - \mathbf{A}_1\eta\|_2^2 + \bar{\mathcal{L}}(\mathbf{A}_1) \ \text{ s.t. } \ \|\xi_{\mathbf{A}_1}\|_{\infty,\infty} \leq \epsilon_2.$$

The previous problem can be solved with proximal gradient method.

# F    EXPERIMENTAL DETAILS AND MORE RESULTS

In this section, we are going to describe the settings and the values of the hyper-parameters that we used in the experiments. Moreover, we will show more results since we have limited space in the main paper.

## F.1    TROPICAL VIEW TO THE LOTTERY TICKET HYPOTHESIS.

We begin by throwing the following question. *Why investigating the tropical geometrical perspective of the decision boundaries is more important than investigating the tropical geometrical representation of the functional form of the network ?* In this section, we show one more experiment that differentiate between these two views. In the following, we can see that variations can happen to the tropical geometrical representation of the functional form (zonotopes in case of single hidden layer neural network), but the shape of the polytope of the decision boundaries is still unchanged and consequently, the decision boundaries. For this purpose, we trained a single hidden layer neural network on a simple dataset like the one in Figure 2, then we do several iteration of pruning, and visualise at each iteration both the polytope of the decision boundaries and the zonotopes of the functional representation of the neural network. It can be easily seen that changes in the zonotopes may not change the shape of the decision boundaries polytope and consequently the decision boundaries of the neural network.

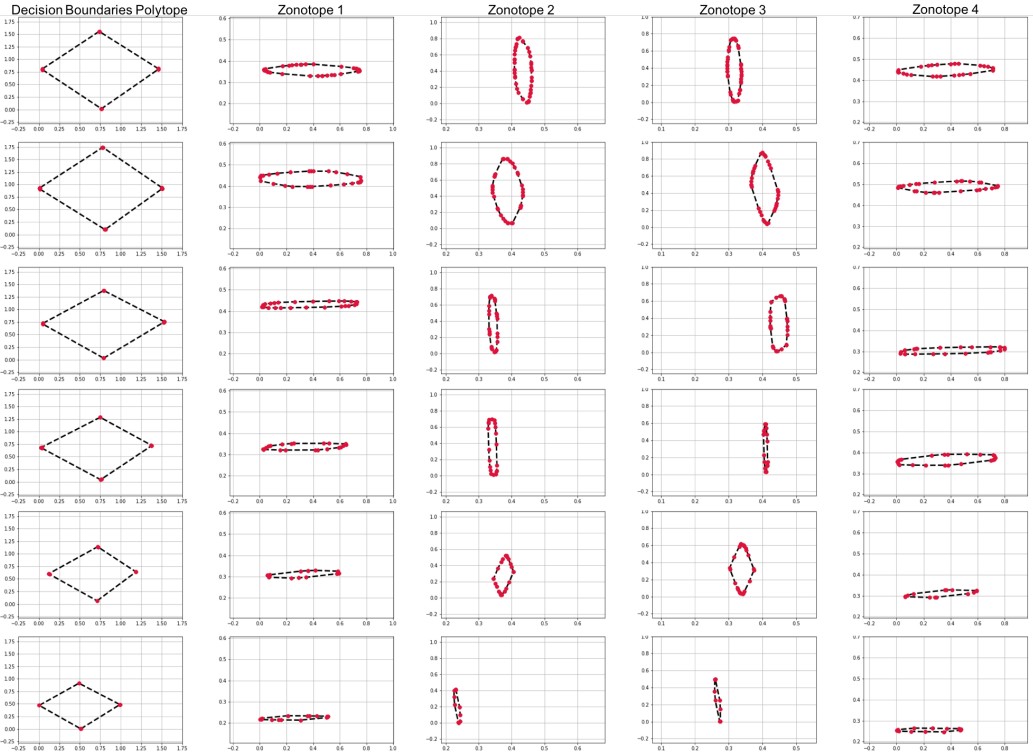

Figure 7: **Changes in Functional Zonotopes and Decision Boundaries Polytope.** First column: decision boundaries polytope, rest of the columns are the geometrical representation of the functional form of the network. Under different pruning iterations using class blind, we can spot the changes that affected the tropical geometric representation of the functional form of the network (zonotopes) while the shape of the decision boundaries polytope is unaffected.

And thus it can be clearly seen that our formulation, which is looking at the decision boundaries polytope is more general, precise and indeed more meaningful.

Moreover, we conducted the same experiment explained in the main paper of this section on another dataset to have further demonstration on the favour that the lottery ticket initialization has over other

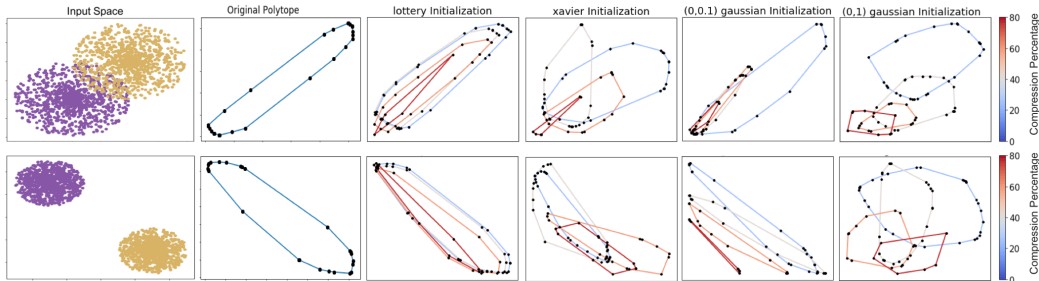

Figure 8: **Effect of Different Initializations on the Decision Boundaries Polytope.** From left to right: training dataset, decision boundaries polytope of original network followed by the decision boundaries polytope during several iterations of pruning with different initializations.

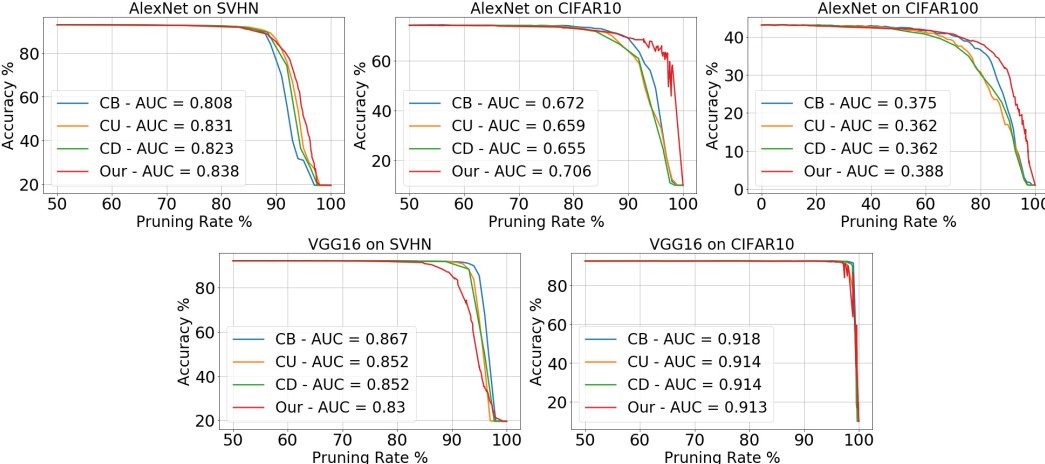

Figure 9: **Effect of Different Initializations on the Decision Boundaries Polytope.** From left to right: training dataset, decision boundaries polytope of original network followed by the decision boundaries polytope during several iterations of pruning with different initializations.

initialization when pruning and retraining the pruned model. It is clear that the lottery initializations is the one that preserves the shape of the decision boundaries polytope the most.

### F.2 TROPICAL PRUNING

In the tropical pruning, we have control on two hyper-parameters only, namely the number of iterations and the regularizer coefficient $\lambda$ which controls the pruning rate. In all of the experiments, we ran the algorithm for 1 iteration only and we increase $\lambda$ starting from $0.02$ linearly with a factor of $0.01$ to reach $100\%$ pruning. It is also worthy to mention that the output of the algorithm will be new sparse matrices $\tilde{A}, \tilde{B}$, but the new network parameters will be the elements in the original matrices $A, B$ that have indices correspond to the indices of non-zero elements in $\tilde{A}, \tilde{B}$. By that, the algorithm removes the non-effective line segments that do not contribute to the decision boundaries polytope, without changing the non-deleted segments. Above all, more results of pruning of AlexNet and VGG16 on various datasets are shown below.

Figure 10: **Results of Tropical Pruning with Fine Tuning the Biases of the Classifier.** Tropical pruning applied on AlexNet and VGG16 trained on SVHN, CIFAR10, CIFAR100 against different pruning methods with fine tuning the biases of the classifier only.

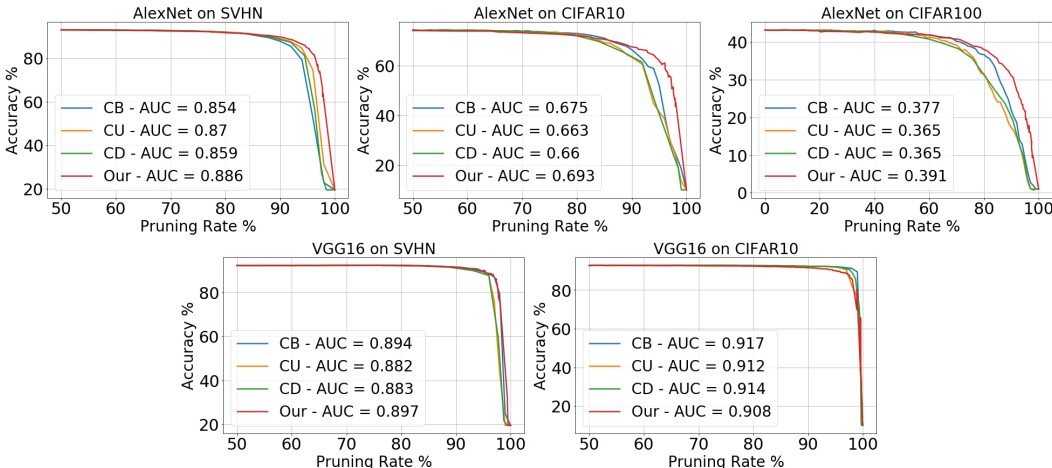

Figure 11: **Results of Tropical Pruning with Fine Tuning the Biases of the Network.** Tropical pruning applied on AlexNet and VGG16 trained on SVHN, CIFAR10, CIFAR100 against different pruning methods with fine tuning the biases of the network.

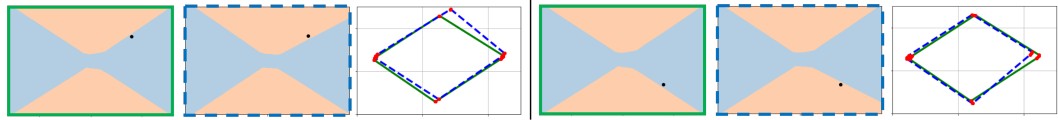

Figure 12: **Dual View of Tropical Adversarial Attacks.** Effect of tropical adversarial attack on a synthetic dataset with two classes in two different scenarios for the black input point. From left to right: decision boundaries of Original model, perturbed model and decision boundaries polytopes(green for original model and blue for perturbed model).

## F.3 TROPICAL ADVERSARIAL ATTACK

For the tropical adversarial attack, we control five different hyper parameters which are

$\epsilon_1$ : The upper bound for the infinite norm of $\delta$.

$\epsilon_2$ : The upper bound for the $\|.\|_{\infty,\infty}$ of the perturbation on the first linear layer.

$\lambda$ : Regularizer to enforce the equality between input perturbation and first layer perturbation

$\eta$ : Bergman divergence constant.

$\rho$ : ADMM constant.

For all of the experiments, $\{\epsilon_2, \lambda, \eta, \rho\}$ had the values $\{1, 10^{-3}, 2.5, 1\}$ respectively. the value of $\epsilon_1$ was $0.1$ when attacking the -fours- images, and $0.2$ for the rest of the images. Finally, we show extra results of attacking the decision boundaries of synthetic data in $\mathbb{R}^2$ and MNIST images by tropical adversarial attacks.

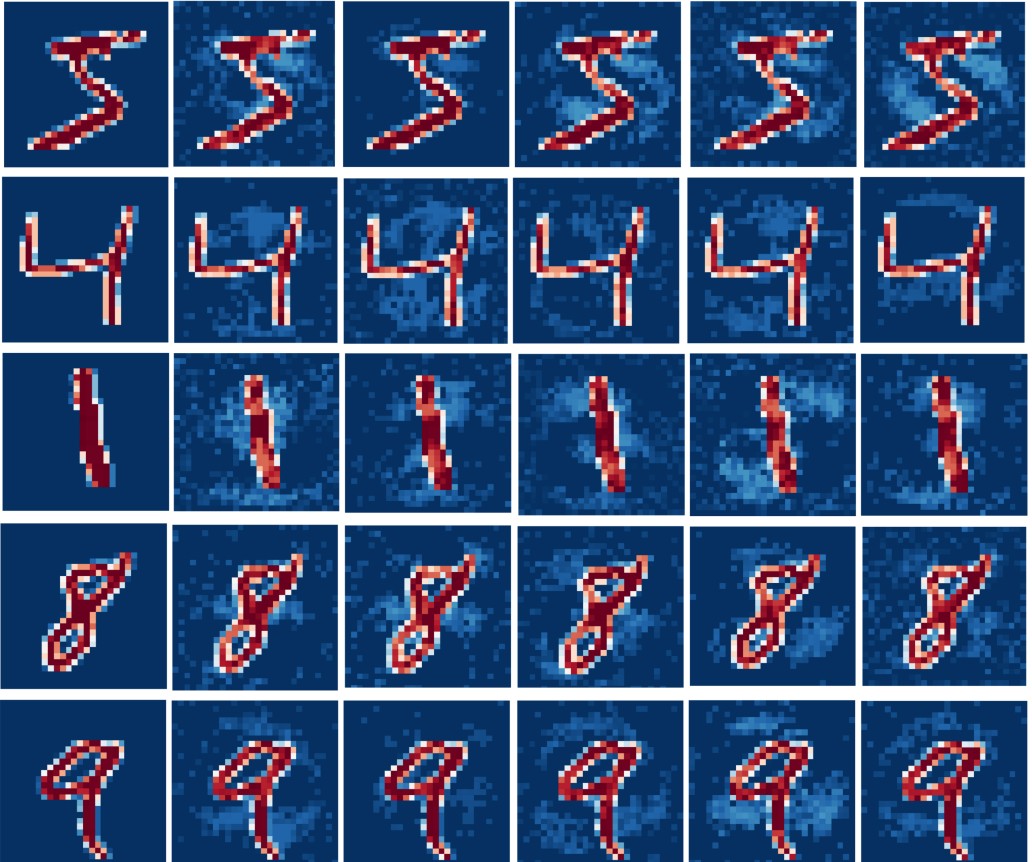

Figure 13: **Effect of Tropical Adversarial Attacks on MNIST Images.** First row from the left: Clean image, perturbed images classified as [7,3,2,1,0] respectively. Second row from left: Clean image, perturbed images classified as [9,8,7,3,2] respectively. Third row from left: Clean image, perturbed images classified as [9,8,7,5,3] respectively. Fourth row from left: Clean image, perturbed images classified as [9,4,3,2,1] respectively. Fifth row from left: Clean image, perturbed images classified as [8,4,3,2,1] respectively.

# G  REBUTTAL

## G.1  REVIEWER 3

We thank R3 for the time spent reviewing the paper. It is though not clear to the authors the main reason behind the initial score of weak reject as R3 seems to have very generic questions about our work but not a particular criticism of the novelty/contribution that we can address in our rebuttal. We hope that the following response addresses and clarifies some key elements. Moreover, we want to bring to the attention of R3 that we have addressed the comments/typos/suggestions of all reviewers in the revised version and marked them in blue.

**Q1: What benefit does introducing tropical geometry brings in terms of theoretical analysis? Does using tropical geometry give us the theoretical results that traditional analysis can not give us? If so, what is it? I am trying to understand why the authors use this tool. The authors should be explicit in their motivation so that the readers are clear about the contribution of this paper. More specifically, from my perspective, tropical semiring, tropical polynomials and tropical rational functions all can be represented with the standard mathematical tools. Here they are just redefining several concepts.**

As discussed thoroughly in the introduction (last paragraph of page 1), tropical geometry is the younger twin to algebraic geometry on a particular semiring defined in a way to align with the study

of piecewise linear functions. The early definitions stated in the paper (1 to 5) are well known in the TG literature and were restated for the completion of this paper. While it is true that the definitions can be represented with standard mathematical tools; however, this misses the fundamental powerful element TG promises. TG transforms algebraic problems of piecewise linear nature to a combinatoric problem on general polytopes. To that end, Zhang et. al. 2018 (to the best of our knowledge the only work at the intersection between TG and DNNs) rederived classical results (upper bound on the number of linear pieces of DNNs) in a much simpler analysis by counting vertices on polytopes. In this work, instead of studying the functional representation of piecewise linear DNNs, we study their decision boundaries using the lens of TG. To wit, the geometric characterization of the decision boundaries of DNNs developed in Theorem 2 cannot be attained using standard mathematical tools. More specifically, Theorem 2 represented a superset to the decision boundaries (the tropical hypersurface $\mathcal{T}(R(\mathbf{x}))$, with a geometric structure that is the convex hull between two zonotopes. While this by itself opens doors for a family of new geometrically motivated regualrizers for training DNNs that are in direct correspondence with the behaviour of the decision boundaries, we do not dwell on training beyond this point and leave that for future work. However, this new result allowed for re-affirmation to the lottery ticket hypothesis in a new fresh perspective. Moreover, we propose new optimization problems that are geometrically motivated (based on Theorem 2) for several classical problems, i.e. network pruning and adversarial attacks that were not possible before and have provided several new insights and directions. That is we show an intimate relation between network perturbations (through decision boundary polytope perturbations) and the construction of adversarial attacks (input perturbations).

**Q2: In Experiments on Tropical Pruning, the authors mentioned we compare our tropical pruning approach against Class Blind (CB), Class Uniform (CU), and Class Distribution (CD) methods Han et al. (2015). What is Class Blind, Class Uniform and Class Distribution? There seems to be an error here Figure 5 shows the pruning comparison between our tropical approach ..., i think Figure 5 should be Figure 4.**

We have added the definition of the pruning methods of Han et al. (2015) in the revised version of the paper for completion, and corrected the typo in the Figure reference.

**Q3: In the adversarial attack part, is the authors proposing a new attack method? If so, then the authors should report the test accuracy under attack. Also, the experimental results should not be restricted to MNIST dataset. I am also not sure about the attack settings here, the authors said Instead of designing a sample noise such that (x0 + $\eta$) belongs to a new decision region, one can instead fix x0 and perturb the network parameters to move the decision boundaries in a way that x0 appears in a new classification region.. Why use this setting? Are there any intuitions? Since this is different from traditional adversarial attack terminology, the authors should stop using adversarial attacks as in tropical adversarial attacks because it is really misleading.**

As highlighted in the last sentence in section 6, we are not competing against other attacks, but we rather show how this new geometric view to the decision boundaries provided by the TG analysis in Theorem 2 can be leveraged for the construction of adversarial attacks. We want to emphasize to R3 that the polytope representing the decision boundaries (convex hull of two zonotopes as per Theorem 2) is a function of the network parameters and not of the input space. Thus, it is not initially clear how one can frame the adversarial attacks problem in this new fresh tropical setting since adversarial attacks is the task of perturbing the input space as opposed to the parameters space of the network resulting in a flip in the prediction. In the tropical adversarial attacks section, we show that the problem of designing an adversarial attack $\mathbf{x}_0 + \eta$ that flips the network prediction is closely related to the problem of flipping the network prediction by perturbing the network parameters in the first layer $\mathbf{A}_1 + \zeta_{\mathbf{A}_1}$ where both problems are related through a linear system. That is to say, if one finds $\zeta_{\mathbf{A}_1}$ that perturbs the geometric structure (convex hull between two zonotopes, i.e. decision boundaries) sufficiently enough to flip the network prediction, one can find an equivalent pixel adversarial attack $\eta$ by solving the linear system $\mathbf{A}_1 \eta = \zeta_{\mathbf{A}_1} \mathbf{x}_0$ that flips the prediction of the original unperturbed network (see the end of page 7). We thereafter propose Problem (5) incorporating the geometric information from Theorem 2 where the linear system is accounted for in the constraints set. We propose an algorithm to solve the problem (a mix of penalty and ADMM) detailed in Algorithm 1 in the appendix. The solution to problem 5 by applying Algorithm 1 results in the construction of adversarial attacks ($\eta$) that indeed flip the network prediction over all tested examples on the MNIST dataset.

## G.2 REVIEWER 2

We thank R2 for the time spent reviewing the paper. We also thank R2 for acknowledging our technical and theoretical contributions. Please note that we have addressed the comments/typos/suggestions of all reviewers in the revised version and marked them in blue. Follows our response.

**Q1: This paper needs to be placed properly among several important missing references on the decision boundary of deep neural networks [1][2]. In particular, using introduced tropical geometry perspective, how we can obtain the complexity of the decision boundary of a deep neural network?**

The two works referenced by R2 are not directly related to the body of our work. Below, we summarize both works and state how our work is vastly different from both. The authors of [1] show that under certain assumptions, the decision boundaries of the last fully connected layer converges to an SVM classifier. That is to say, the features learnt in deep neural networks are linearly separable with max margin type linear classifier. On the other hand, the authors of [2] showed that the decision regions of neural networks with width smaller than the input dimension are unbounded. In our work, we use a new type of analysis (tropical geometry) to represent the set of decision boundaries $\mathcal{B}$ through its superset $\mathcal{T}(R(\mathbf{x}))$ that is the solution set to the tropical polynomial $R(\mathbf{x})$. We then show that this solution set is related to a geometric structure referred to as the decision boundaries polytope (convex hull between two zonotopes), this is analogous to constructing newton polytopes for the solution sets to classical polynomials in algebraic geoemtry. The normals to the edges of this polytope are parallel to the superset of the decision boundaries $\mathcal{T}(R(\mathbf{x}))$. That is to say, if one processes the polytope in an way that preserves the direction of the normals, the decision boundaries of the network are preserved. This is the base idea behind all later experiments. In general, this new representation presents a new fresh revisit to the lottery ticket hypothesis and an utterly new view to network pruning and adversarial attacks. We do believe this new representation can be of benefit to other applications and can open doors for a family of new geometrically inspired network regularizers as well.

**Q2: Regarding the complexity of the decision boundaries of neural networks.**

The only work at the intersection between TG and DNNs that we are aware of is the work of Zhang et. al. 2018. They re-derived a classical upper bound to the number of linear regions of feed forward neural networks with ReLU activations by counting vertices of polytopes (Theorem 6.3). The work was limited to the complexity (number of linear regions) of the functional representation of the network and not to the decision boundary.

**Q3: The second part of Theorem 2 should be explained straightforwardly and clearly as it plays an important role in the subsequent results and applications.**

We have added a "Digesting Theorem 2" paragraph in the revised version and rearranged the structure a bit around Theorem 2.

**Q4: Pruning convolutional layers.**

Most of the parameters (memory complexity) are the in fully connected layers. For example, the convolutional part of VGG16 has 14,714,688 parameters, whereas the fully connected layers have 262,000,400 parameters in total which is 17 times larger. Similarly, the convolutional part of AlexNet has 3,747,200 parameters while only the first fully connected layer has 37,752,832 parameters [5]. However, efficiently extending the tropical pruning to convolutional layers is a nontrivial interesting direction. Generally speaking, convolutional layers fit our framework naturally since a convolutional kernel can be represented with a structured topelitz/circulant matrix. However, a question of efficiency still remains as one still needs to construct the underlying structured matrix representing the convolutional kernel. Thereafter, a direction of interest is the tropical formulation of the network pruning problem as a function of the convolutional kernels surpassing the need for the construction of the dense representation of the kernel. We keep this for future work.

**Q5: For the similarity measure.**

Comparing the exact decision boundaries between two different architectures can be very difficult in the sense where decision boundaries for a two-class output network $f$ are defined as $\{\mathbf{x} \in \mathbb{R}^n : f_1(\mathbf{x}) = f_2(\mathbf{x})\}$. Another approach to compare decision boundaries, which is proposed by our work, is by computing the distance between the the dual subdivision polytope $(\delta(R(\mathbf{x})))$ of the tropical

polynomials $R(\mathbf{x})$ representing two different architectures. This is since the normals to the edges of the polytope $(\delta(R(\mathbf{x})))$ are parallel to a superset of the decision boundaries (see Figure 1). This is exactly the proposed objective (1) where $d(.)$ is a distance function to compare the orientation between two general polytopes. Since finding a good choice for $d(.)$ is generally difficult we instead approximate it by comparing the generators constructing the $(\delta(R(\mathbf{x})))$ in Euclidean distance for ease (objective (2)). Experimentally and following prior art in the pruning literature (Han et. al. 2015), to compare the effectiveness of the pruning scheme, we compare the test accuracies across architectures as a function of the pruning ratio. Regardless, the reviewer is right about that similar test accuracies does not imply similar decision boundaries but rather only an indication.

**Q6: In adversarial examples generation, typically for a pre-trained deep neural network model one is interested in generating examples that are misclassified by the model while they resemble real instances. In this setting, we keep the model and thus its decision boundary intact. In this paper, nevertheless, aiming at generating adversarial examples, the decision boundary and thus the (pre-trained) model is altered. By chaining the decision boundary, however, the model's decisions for original real samples might change as well. Therefore, it is not clear to the reviewer how the introduced method is comparable to the well-established adversarial example generation setting.**

The new approach is definitely comparable to the well-establisehd adversarial example generation setting. Let us explain. The new analysis provided by Theorem 2, allows to present the decision boundaries geometrically as a convex hull between two zonotopes which is a function of only the network parameters and not the input space. Thus, it not clear how one can frame the adversarial attacks problem in this new fresh tropical setting since adversarial attacks is the task of perturbing the input space as opposed to the parameters space of the network resulting in a flip in the prediction. In the tropical adversarial attacks section, we show that the problem of designing an adversarial attack $\mathbf{x}_0 + \eta$ that flips the network prediction is closely related to the problem of flipping the network prediction by perturbing the network parameters in the first layer $\mathbf{A}_1 + \zeta_{\mathbf{A}_1}$ where both problems are related through a linear system. That is to say, if one finds $\zeta_{\mathbf{A}_1}$ (perturbations in the first linear layer) that perturbs the geometric structure (convex hull between two zonotopes, i.e. decision boundaries) sufficiently enough to flip the network prediction, one can find an equivalent pixel adversarial attack $\eta$ by solving the linear system $\mathbf{A}_1 \eta = \zeta_{\mathbf{A}_1} \mathbf{x}_0$ that flips the prediction of the original unperturbed network (see the end of page 7). We incorporate this in an overall objective (5) where the linear system is incorporated as a constraint. Upon solving (5) with the proposed Algorithm 1, we attack the original network (unperturbed) with the adversarial attack $\eta$. Therefore, this is comparable to the classical adversarial attacks framework. This approach indeed resulted into flipping the network prediction over all tested examples on the MNIST dataset. As highlighted at the end of section 6, we do not aim in our approach to outperform the state of the art adversarial attacks, but rather to provide a novel geometrically inspired perspective that can shed new light in this field.

**Q7: Two previous papers investigated the decision boundary of the deep neural networks in the presence of adversarial examples [3][4]. Please discuss how the introduced method in this paper is placed among these methods.**

The work of [3] analyzed the geometry of adversarial examples by means of manifold reconstruction to study the trade off between robustness under different norms. On the other hand, [4] crafted adversarial attacks by estimating the distance to the decision boundaries using random search directions. Both of the papers made a local estimation of the decision boundary around the attacked point to construct the adversarial attack to the input image. In our work, we geometrically characterized the decision boundaries in Theorem 2 where the polytope (convex hull of two zonotopes) is only a function of the network parameters and NOT the input space. We presented a dual view to adversarial attacks in which one can construct adversarial examples by investigating network parameters perturbations that results in the largest perturbation to this polytope representing the decision boundaries. The scope of our work, and unlike prior art [3,4], is focused towards a new geometric polytope representation of the decision boundary in the network parameter space (not the input space) through a new novel analysis. We have added a discussion of both papers in the adversarial attacks section (Section 6).

**Minor comments.**

We have addressed the concerns of R2 and left the changes in blue in the revised version.

[1] "On the decision boundary of deep neural networks". SLi, Yu and Richtarik, Peter and Ding, Lizhong and Gao, Xin.

[2] "On decision regions of narrow deep neural networks". Beise, Hans-Peter and Da Cruz, Steve Dias and Schröder, Udo.

[3] "On the geometry of adversarial examples". Marc Khoury and and Dylan Hadfield-Menell.

[4] "Decision Boundary Analysis of Adversarial Examples". Warren He, Bo Li and Dawn Song.

[5] "Learning both weights and connections for efficient neural networks". Song Han, Jeff Pool, John Tran, and William J. Dally.

### G.3   REVIEWER 1

We thank R1 for the constructive detailed thorough review of the paper and for acknowledging our contributions and the new insights. Follows our response to R1's concerns.

**In regards to clarity, exposition and focus.**

To improve the clarity and exposition, we have added several paragraphs in the revised version of the paper. The revised edits are marked in blue. As in regards to the focus, we found that it is challenging within a reasonable time to carry out this major change in the paper. To that end, we have done our best to further elaborate on several key results in the paper. For instance, we have merged the paragraph above the contributions with the contributions paragraph. We have added another paragraph dissecting Theorem 2. We have added some few relevant references that are essential for the context of motivating tropical adversarial attacks.

**In regards to the suggestions.**

• **Adding the information that the semiring lacks the additive inverse.**

This has been addressed in the revised version.

•**Adding tropical quotient to definition 1.**

This has been addressed in the revised version.

• **Definition of $\pi$ and the upper faces.**

Indeed, $\pi$ is a projection operator that drops the last coordinate. As for upper faces, the formal definition is given as follows, for a polytope $P$, $F$ is an upper face of $P$ if $\mathbf{x} + t\mathbf{e} \notin P$ for any $\mathbf{x} \in F, t > 0$ where $\mathbf{e}$ is a canonical vector. That is the faces that can be seen from "above". A good graphical example can be found in Figure 2 from Zhang et. al. 2018. We dropped this definition from the paper as it may add some confusion while it does not play an important role in the later analysis.

• **Theorem 2 lacks intuitive formulation.**

This has been addressed in the revised version and we have rearranged the structure of text below Theorem 2.

• **Regarding the issue of the tropical hypersurface.**

Indeed, the superset is in terms of set theory. That is to say, the set of decision boundaries $\mathcal{B}$ is a supset of the tropical hypersurface set $\mathcal{T}(R(\mathbf{x}))$.

• **Regarding Figure 2.**

The color map represents the compression percentage. We have added a legend to Figure 2. Note that the second figure in Figure 2 tilted "original polytope" represents the polytope of the dual subdivision (convex hull between two zonotopes). While the polytope seems to have only 4 vertices, there are in fact many other overlapping vertices and thereafter many small edges between all the seemingly overlapping vertices. It is to observe that the normals to only the 4 major edges in the "original polytope" are indeed parallel to the decision boundaries plotted by performing forward passes through the network in the first figure titled "Input Space". Note that despite the fact that more compression is performed, the orientation of the overall polytope is preserved for the lottery ticket initialization and thereafter preserving the main orientation of the decision boundaries. This is unlike the the other types of initialization where the orientation of the polytope is vastly different with different compression ratios resulting into a larger change in the orientation of the decision boundaries.

• **I would suggest to place Figure 1 after stating Theorem 2, since it is only referenced later on. Furthermore, the red structures are somewhat confusing. According to Theorem 2, the decision boundary is a subset of the hypersurface, right? What is the relation of the red structures in the convex hull visualisation? The caption states that they are normals, but as far as I can tell, this has not been formalised anywhere in the paper (it is used later on, though).**

Correct. The decision boundaries are subsets of the tropical hypersurfaces. However, it has been shown by Maclagan & Sturmfels (2015) (Propositon 3.1.6) that the tropical hypersurface $\mathcal{T}$ to any d variate tropical polynomial is the (d-1)-skeleton of the polyhedral complex dual to the dual subdivision $\delta()$. This implies that the normals to the edges (faces in higher dimensions) are parallel to the tropical hypersurface. If R1 is interested in learning more about this, an excellent starting point to build up this intuition is the work of Erwan Brugall and Kristin Shaw "A bit of tropical geometry". Please refer to section 2.2 "Dual subdivisions" pages 6 and 7.

• **The discussion about the functional form.**

We elaborate on this in section F in the appendix. Instead of investigating the decision boundaries polytope of the tropical polynomial representing the decision boundaries $R(\mathbf{x})$, we analyze the 4 different tropical polynomials representing the 2 different classes. Recall each output function of the network is a tropical rational of two tropical polynomials giving rise to 4 different polytopes. Figure 7 shows that the decision boundary polytope $\delta(R(\mathbf{x}))$ with the lottery ticket is hardly changed while pruning the network (first column in Figure 7). On the contrary, the zonotopes (dual subdivisions of the 4 different tropical polynomials) vary much more significantly (coloumns 2,3,4,5). This demonstrates that there can exist different tropical polynomials representing the functional form of the network (*i.e.* $H_{1,2}$ and $Q_{1,2}$) while having the same structure for the decision boundary polytope the corresponding first figure in the same row of Figure 7. This is a mere observation and worth investigating in future work.

• **In Section 4, how many experiments of the sort were performed? I find this a highly instructive view so I would love to see more experiments of this sort. Do these claims hold over multiple repetitions and for (slightly) larger architectures as well?**

We already had one extra experiment (Figure 8) in the appendix for another data. We have also based on the suggestion of R1 added two more experiments (Figure 9) on two other datasets. Note that extracting the decision boundaries polytope for larger architectures is much more difficult for two different reasons. First, for deeper networks beyond the structure Affine-ReLU-Affine, the decision boundaries polytope is generic and no longer enjoys the nice properties zonotopes exhibit. Enumerating their vertices rapidly turns to a computationally intractable problem. Secondly, one can perhaps visualize the polytope for networks with 3 dimensional input but it gets trickier beyond that.

• **Regarding that the claim that orientations are preserved should be formalised.**

That is an excellent question. We have investigated several metrics that try to capture information about the orientation of a given polytope. For instance, we have investigated the feature that is the histogram of the oriented normals. That is a histogram of angles for all the normals to edges given a polytope. We have also investigated the Hausdorff distance as a metric between polytopes. However, we have decided to keep this for a future direction as this is by itself a entirely new line of work. That is designing the distance functions $d(.)$ between polytopes that captures orientation information that can be used for tropical pruning (objective 8) or perhaps other applications. Another interesting direction is whether such a distance function can be learnt in a meta-learning fashion. For now, we restrict the experiments to the approximation used in objective (2) which for now only captures the distance between the sets of generators of the zonotopes in Eucledian sense.

• **Adding link to the definition of minkowski sum in the appendix.**

This has been addressed in the revised version.

• **Description to other pruning methods.**

This has been addressed in the revised version where we have added the definition to all pruning competitors.

• **About the plots in Figure 4.**

In the experiments of Figure 4, there is no stochasticity. All base networks (AlexNet and VGG16) are trained before hand to achieve the state-of-art baseline results on the respective datasets (SVHN, CIFAR10 and CIFAR100). The networks are then fixed and several pruning schemes, including the tropical approach, are applied with a varying level of pruning ratio. We do not re-train the networks after each pruning step. Thus there exists no source of randomness. However, this is still an excellent observation. This is since we conduct experiments in the appendix where after each pruning ratio we fine tune only the biases of the classifier (Figure 10) or we fine tune the biases of the complete network (Figure 11). In such experiments, due to the fine tuning step, we will consider in the final version to report the results averaged over multiple runs.

- **In Section 6, I find the comment on normals generating a superset to the decision boundaries hard to understand.**

Indeed, the wording of the sentence was sub optimal. The statement was to reassure what has been established in earlier sections that the normals to the decision boundary polytope $\delta(R(\mathbf{x}))$ represent the tropical hypersurface set $\mathcal{T}(R(\mathbf{x}))$ which is a super set to the decision boundaries set $\mathcal{B}$ as per Theorem 2. We have rephrased the sentence.

- **Take-away message regarding perturbing the decision boundaries.**

The proposed approach of perturbing the decision boundaries is a mere dual view for adversarial attacks. That is to say, to flip a network prediction for a sample $\mathbf{x}_0$, one can either adversarially perturb the sample (add noise) to cross the decision boundary to a new classification region or one can perturb the decision boundaries to move closer to the sample $\mathbf{x}_0$ to appear as if it is in a new classification region. Figure 5, demonstrates the later visually through perturbing the decision boundary polytope. One can observe that perturbing the correct edge in the polytope in Figure 5, corresponds to altering a specific decision boundary. Figure 5, indeed as correctly pointed out of R1, is a feasibility study showing that one can perturb decision boundary by perturbing the dual subdiviosn polytope. However, the real take away message is that these two views (perturbing the input or the parameter space) are intimately related through a linear system as discussed in the subsection titled "Dual View to Adversarial Attacks". We propose an objective function (Problem 5) and an algorithm (Algorithm 1) to tackle this problem. The solution to problem (5) provides an INPUT perturbation that results in altering the network prediction. That is to say, the new framework allows for incorporating geometrically motivated objective function towards constructing classical adversarial attacks.

- **Regarding the future extension on CNNs and GCNs.**

We are currently investigating efficient extension of the current results to convolutional layers. Note that while convolutional layers can be represented with a large structured topelitz/circulant matrix, we are interested in extensions that allow for similar analysis but as a function of the convolutional kernel surpassing the need to constructing convolutional matrix. The GCNs is definitely an exciting excellent future direction that we have not yet entertained.

- **Minor style issues.** We have addressed all the style issues in the revised version.

