# OpenReview forum: "On the Decision Boundaries of Deep Neural Networks: A Tropical Geometry Perspective"
_ICLR.cc/2020/Conference — Reject_

### Official Review · AnonReviewer1 · 2019-10-23
**Official Blind Review #1**

**Rating:** 8

**Review:**

1. Summary of the paper

This paper describes the decision boundaries of a certain class of
neural networks (piecewise linear, non-linear activation functions)
through the lens of tropical geometry.
An earlier result by Zhang et al. (2018) is extended to multi-class
classification problems (technically, only the result for a binary
classification is given in the main text, though).

Similar to this earlier work, the network is shown to be represented as
a tropical rational function. The dual subdivision of this function is
shown to be represented as the convex hull of two zonotopes.

This characterisation is used to explain different phenomena of neural
network training, viz. the 'lottery ticket hypothesis', network
pruning, and adversarial attacks.

2. Summary of the review

This is a highly interesting paper with a very relevant subject. I think
that the perspective of tropical geometry leads to valuable insights. My
background is *not* in tropical geometry, so this paper required several
passes to fully grasp.

I like the novel insights that this paper creates; it is very interesting
to observe known phenomena via tropical geometry. I appreciate the
thorough description of all concepts in this paper. This is to some
extent both 'boon and bane': on the one hand, the paper contains a lot
of information and concepts that need to be understood; on the other
hand, the experiments go *wide* but not *deep*. I suggest to accept the
paper, but to fully endorse it, I would recommend to work on the
following issues:

- Clarity & exposition: In some places, the paper could build intuition
  for non-experts (such as myself) better. This is closely tied to the
  second point.

- Focus: I would maybe pick *one* or *two* of the experimental areas and
  use the remaining space to explain all concepts in more detail, build
  some intuition, and provide a more in-depth setup. Nothing has to
  removed of course; it could still be put in the appendix.

This paper has the potential to be a very strong insightful contribution
to to our community; the authors are to be commended!

3. Detailed comments (clarity)

The paper describes its concepts well and has a high information
density. At times, there is the risk that readers are provided with too
much information in the main text, leaving the necessary intuition
somewhat lacking (unless they are already experts in the subject matter,
in which case a lot of the information can be skipped).

I realise that writing a paper based on methods that are not yet
well-established is no small feat; the authors are to be commended for
that!

Here are some suggestions from someone with a background in differential
topology:

- The introduction and contributions are somewhat repetitive; I would
  suggest merging the 'Contributions.' paragraph with the one preceding
  it

- Even though it *should* be a well-known definition, I would briefly
  explain that the semiring lacks an additive inverse

- Add an explanation of the tropical quotient to Definition 1; I find
  the current phrasing of Definition 3 to be confusing at first glance

- What are *upper faces*? Faces with a specific coordinate fixed?

- A definition of $\pi$ is required. It is my understanding that $\pi$
  is a projection function that 'drops' the last coordinate. Is this
  correct? If so, it should be briefly mentioned on p. 3; else, the
  discussion about the bias-free case on p. 4 cannot be understood.

- Theorem 2 lacks an 'intuitive' formulation; the results are stated in
  a terse mathematical fashion, but it would be helpful (also in light
  of the subsequent discussion) to briefly comment on their *meaning*.

  For example, the first result could be restated as 'the decision
  boundary is a subset of a tropical hypersurface of the polynomial
  $R(x)$'.

  The paragraph 'Theorem 2 bridges the gap...' could maybe also be moved
  to _precede_ the theorem statement.

- The paper refers to $\mathcal{T}(R(x))$ as a super-set, but in my
  understanding, it is a *level set* because Definition 4 uses an
  equality, not an inequality. Am I misunderstanding this? One potential
  misinterpretation of my part could be that superset refers to the fact
  that $\mathcal{B} \subseteq \mathcal{T}(R(x))$; so not a superset
  in the sense of level set analysis, but rather a superset in terms of
  set theory. If this is the case, maybe rephrase the sentence above to
  something like 'boundaries $\mathcal{B}$ through their superset
  $\mathcal{T}(R(x))$ according to the first statement of Theorem 2'.

- What does the colour map in Figure 2 depict? The number of iterations?
  Moreover, I find the polytope though to understand at first glance;
  how is it related to the decision boundaries that are shown in the
  leftmost figure?

- I would suggest to place Figure 1 after stating Theorem 2, since it is
  only referenced later on. Furthermore, the red structures are somewhat
  confusing. According to Theorem 2, the decision boundary is a subset
  of the hypersurface, right? What is the relation of the red structures
  in the convex hull visualisation? The caption states that they are
  normals, but as far as I can tell, this has not been formalised
  anywhere in the paper (it is used later on, though).

- To what extent is the existence of the functions described by Theorem 2
  unique? On p. 5, in Section 4, the paper alludes to *not* using the
  functional form of the network directly because it does not seem to be
  unique. I would like this to be explained a in more details, as
  I found the justification of why the dual subdivision is used quite
  hard to follow.

- In Section 4, how many experiments of the sort were performed? I find
  this a highly instructive view so I would love to see more experiments
  of this sort. Do these claims hold over multiple repetitions and for
  (slightly) larger architectures as well?

  Please also see my comments on Figure 2 above.

- The claim that orientations are preserved should be formalised.
  I immediately understand the intuition behind this concept, but
  if possible, I would like a quantification of this. Might it be
  possible to *measure* changes in orientation with respect to an
  original polytope? If so, it should be possible to provide more
  experiments about these effects and summarise them accordingly.
  Maybe it would also be interesting to investigate whether other
  initialisations can be compared in terms of their orientations?

- In Section 5, I would give a brief link to the appendix for the
  definition of a Minkowski sum.

- Section 5 has (in contrast to the other sections) a lot of details
  containing the experimental setup, but it is missing a description of
  the competitor methods. Adding to what I wrote above, I feel that the
  paper should rather pick *one* area in which experiments are
  performed; the pruning (together with the lottery ticket hypothesis
  explanation, which could be seen as a motivating example) strikes me
  as a good candidate for this.

  I really like this concept of tropical pruning, by the way---it is an
  elegant, principled description!

- The plots in Figure 4 should summarise multiple pruning runs, if
  possible. Why not show a standard deviation as well? Given the
  stochasticity of training, I would think this highly necessary.

- In Section 6, I find the comment on normals generating a superset to
  the decision boundaries hard to understand.

-  The perspective of perturbing the network such that the decision
   boundaries change is really interesting, but I am missing
   a 'take-away message' or a discussion of the insights. Currently,
   this section seems more like a feasibility study: it appears to be
   possible to use the tropical description to find new parameters that
   misclassify a given input. I would propose a discussion of the
   implications of these findings.

- Concerning future extensions of this work, are there some promising
  results or directions for CNNs or GCNs? If so, it would strengthen
  the conclusion if they were mentioned.

4. Minor style issues

The paper is well-written. I found some minor style issues:

- 'piece-wise' --> 'piecewise' (occur multiple times)
- 'recently demonstrated' --> 'demonstrated'
- 'Thereafter, tropical hypersurfaces divide' --> 'Tropical hypersurfaces divide'
- 'If set $B$ --> 'Letting B'
- I am not sure if I would call adversarial attacks a 'nuisance'; maybe rather a 'problem'?
- Use '\operatorname' or '\mathrm' to typeset the loss in Eq. 5

5. Update after rebuttal

The authors addressed all my important comments; their efforts in rewriting and revising the paper in such a short time period are to be commended. I am very happy to raise my score.

**Experience Assessment:**

I have read many papers in this area.

**Review Assessment: Checking Correctness Of Derivations And Theory:**

I carefully checked the derivations and theory.

**Review Assessment: Checking Correctness Of Experiments:**

I carefully checked the experiments.

**Review Assessment: Thoroughness In Paper Reading:**

I read the paper thoroughly.

---

> ### Author Response · Authors · 2019-11-12
> **Response (1/3) to R1**
>
> We thank R1 for the constructive detailed thorough review of the paper and for acknowledging our contributions and the new insights. Follows our response to R1's concerns.
>
>
> \textbf{In regards to clarity, exposition and focus.}
>
> To improve the clarity and exposition, we have added several paragraphs in the revised version of the paper. The revised edits are marked in blue. As in regards to the focus, we found that it is challenging within a reasonable time to carry out this major change in the paper. To that end, we have done our best to further elaborate on several key results in the paper. For instance, we have merged the paragraph above the contributions with the contributions paragraph. We have added another paragraph dissecting Theorem 2. We have added some few relevant references that are essential for the context of motivating tropical adversarial attacks.
>
> \textbf{In regards to the suggestions.}
>
> $\bullet$ \textbf{Adding the information that the semiring lacks the additive inverse.}
>
> This has been addressed in the revised version.
>
> $\bullet$\textbf{Adding tropical quotient to definition 1.}
>
> This has been addressed in the revised version.
>
> $\bullet$ \textbf{Definition of $\pi$ and the upper faces.}
>
> Indeed, $\pi$ is a projection operator that drops the last coordinate. As for upper faces, the formal definition is given as follows, for a polytope $P$, $F$ is an upper face of $P$ if $\mathbf{x} + t \mathbf{e} \notin P$ for any $\mathbf{x} \in F, t > 0$ where $\mathbf{e}$ is a canonical vector. That is the faces that can be seen from "above". A good graphical example can be found in Figure 2 from Zhang et. al. 2018. We dropped this definition from the paper as it may add some confusion while it does not play an important role in the later analysis.
>
> $\bullet$ \textbf{Theorem 2 lacks intuitive formulation.}
>
> This has been addressed in the revised version and we have rearranged the structure of text below Theorem 2.
>
> $\bullet$ \textbf{Regarding the issue of the tropical hypersurface.}
>
> Indeed, the superset is in terms of set theory. That is to say, the set of decision boundaries $\mathcal{B}$ is a supset of the tropical hypersurface set $\mathcal{T}(R(\mathbf{x}))$.
>
> $\bullet$ \textbf{Regarding Figure 2.}
>
> The color map represents the compression percentage. We have added a legend to Figure 2. Note that the second figure in Figure 2 tilted "original polytope" represents the polytope of the dual subdivision (convex hull between two zonotopes). While the polytope seems to have only 4 vertices, there are in fact many other overlapping vertices and thereafter many small edges between all the seemingly overlapping vertices. It is to observe that the normals to only the 4 major edges in the "original polytope" are indeed parallel to the decision boundaries plotted by performing forward passes through the network in the first figure titled "Input Space". Note that despite the fact that more compression is performed, the orientation of the overall polytope is preserved for the lottery ticket initialization and thereafter preserving the main orientation of the decision boundaries. This is unlike the the other types of initialization where the orientation of the polytope is vastly different with different compression ratios resulting into a larger change in the orientation of the decision boundaries.
>
>
> $\bullet$ \textbf{I would suggest to place Figure 1 after stating Theorem 2, since it is only referenced later on. Furthermore, the red structures are somewhat confusing. According to Theorem 2, the decision boundary is a subset of the hypersurface, right? What is the relation of the red structures in the convex hull visualisation? The caption states that they are normals, but as far as I can tell, this has not been formalised anywhere in the paper (it is used later on, though).}
>
> Correct. The decision boundaries are subsets of the tropical hypersurfaces. However, it has been shown by \cite{maclagan2015tg} (Propositon 3.1.6) that the tropical hypersurface $\mathcal{T}$ to any d variate tropical polynomial is the (d-1)-skeleton of the polyhedral complex dual to the dual subdivision $\delta()$. This implies that the normals to the edges (faces in higher dimensions) are parallel to the tropical hypersurface. If R1 is interested in learning more about this, an excellent starting point to build up this intuition is the work of Erwan Brugallé and Kristin Shaw "A bit of tropical geometry". Please refer to section 2.2 "Dual subdivisions" pages 6 and 7.

---

> > ### Author Response · Authors · 2019-11-12
> > **Response (2/3) to R1**
> >
> > $\bullet$ \textbf{In Section 4, how many experiments of the sort were performed? I find this a highly instructive view so I would love to see more experiments of this sort. Do these claims hold over multiple repetitions and for (slightly) larger architectures as well?}
> >
> > We already had one extra experiment (Figure 8) in the appendix for another data. We have also based on the suggestion of R1 added two more experiments (Figure 9) on two other datasets. Note that extracting the decision boundaries polytope for larger architectures is much more difficult for two different reasons. First, for deeper networks beyond the structure Affine-ReLU-Affine, the decision boundaries polytope is generic and no longer enjoys the nice properties zonotopes exhibit. Enumerating their vertices rapidly turns to a computationally intractable problem. Secondly, one can perhaps visualize the polytope for networks with 3 dimensional input but it gets trickier beyond that.
> >
> >
> > $\bullet$ \textbf{Regarding that the claim that orientations are preserved should be formalised.}
> >
> > That is an excellent question. We have investigated several metrics that try to capture information about the orientation of a given polytope. For instance, we have investigated the feature that is the histogram of the oriented normals. That is a histogram of angles for all the normals to edges given a polytope. We have also investigated the Hausdorff distance as a metric between polytopes. However, we have decided to keep this for a future direction as this is by itself a entirely new line of work. That is designing the distance functions $d(.)$  between polytopes that captures orientation information that can be used for tropical pruning (objective 8) or perhaps other applications. Another interesting direction is whether such a distance function can be learnt in a meta-learning fashion. For now, we restrict the experiments to the approximation used in objective (2) which for now only captures the distance between the sets of generators of the zonotopes in Eucledian sense.
> >
> >
> > $\bullet$ \textbf{Adding link to the definition of minkowski sum  in the appendix.}
> >
> > This has been addressed in the revised version.
> >
> > $\bullet$ \textbf{Description to other pruning methods.}
> >
> > This has been addressed in the revised version where we have added the definition to all pruning competitors.
> >
> > $\bullet$ \textbf{About the plots in Figure 4.}
> >
> > In the experiments of Figure 4, there is no stochasticity. All base networks (AlexNet and VGG16) are trained before hand to achieve the state-of-art baseline results on the respective datasets (SVHN, CIFAR10 and CIFAR100). The networks are then fixed and several pruning schemes, including the tropical approach, are applied with a varying level of pruning ratio. We do not re-train the networks after each pruning step. Thus there exists no source of randomness. However, this is still an excellent observation. This is since we conduct experiments in the appendix where after each pruning ratio we fine tune only the biases of the classifier (Figure 10) or we fine tune the biases of the complete network  (Figure 11). In such experiments, due to the fine tuning step, we will consider in the final version to report the results averaged over multiple runs.
> >
> >
> > $\bullet$ \textbf{In Section 6, I find the comment on normals generating a superset to the decision boundaries hard to understand.}
> >
> > Indeed, the wording of the sentence was sub optimal. The statement was to reassure what has been established in earlier sections that the normals to the decision boundary polytope $\delta(R(\mathbf{x}))$ represent the tropical hypersurface set $\mathcal{T}(R(\mathbf{x}))$ which is a super set to the decision boundaries set $\mathcal{B}$ as per Theorem 2. We have rephrased the sentence.

---

> > > ### Author Response · Authors · 2019-11-12
> > > **Response (3/3) to R1**
> > >
> > > $\bullet$ \textbf{Take-away message regarding perturbing the decision boundaries.}
> > >
> > > The proposed approach of perturbing the decision boundaries is a mere dual view for adversarial attacks. That is to say, to flip a network prediction for a sample $\mathbf{x}_0$, one can either adversarially perturb the sample (add noise) to cross the decision boundary to a new classification region or one can perturb the decision boundaries to move closer to the sample $\mathbf{x}_0$ to appear as if it is in a new classification region. Figure 5, demonstrates the later visually through perturbing the decision boundary polytope. One can observe that perturbing the correct edge in the polytope in Figure 5, corresponds to altering a specific decision boundary. Figure 5, indeed as correctly pointed out of R1, is a feasibility study showing that one can perturb decision boundary by perturbing the dual subdiviosn polytope. However, the real take away message is that these two views (perturbing the input or the parameter space) are intimately related through a linear system as discussed in the subsection titled "Dual View to Adversarial Attacks". We propose an objective function (Problem 5) and an algorithm (Algorithm 1) to tackle this problem. The solution to problem (5) provides an INPUT perturbation that results in altering the network prediction. That is to say, the new framework allows for incorporating geometrically motivated objective function towards constructing classical adversarial attacks.
> > >
> > > $\bullet$ \textbf{Regarding the future extension on CNNs and GCNs.}
> > >
> > > We are currently investigating efficient extension of the current results to convolutional layers. Note that while convolutional layers can be represented with a large structured topelitz/circulant matrix, we are interested in extensions that allow for similar analysis but as a function of the convolutional kernel surpassing the need to constructing convolutional matrix. The GCNs is definitely an exciting excellent future direction that we have not yet entertained.
> > >
> > > $\bullet$ \textbf{Minor style issues.} We have addressed all the style issues in the revised version.

---

> > > > ### Comment · AnonReviewer1 · 2019-11-15
> > > > **re: Your rebuttal**
> > > >
> > > > Thank you very much for addressing my review in this thorough fashion! The paper really benefitted from the additional revision and I really appreciate the work you did here. Thus, I am raising my score.
> > > >
> > > > I have one minor thing to add, which I hope to be seen in a final version of the paper at some point (but it's definitely not necessary in the few hours that remain for the rebuttal!): the decision boundary figures would benefit from some additional clarification. Frist readers of this paper might be look for decision boundaries that are (roughly) orthogonal to the classes, such as the boundary shown in Figure 1. In Figure 2, for example, the decision boundaries polytope is shown. Maybe showing decision boundaries at least for the 'Original polytope' subfigure would make it easier to see the relation between the polytope and the boundary.
> > > >
> > > > That being said, I want to stress that this is a very minor issue and has no bearing whatsoever on my final recommendation.

---

### Official Review · AnonReviewer2 · 2019-10-23
**Official Blind Review #2**

**Rating:** 3

**Review:**

This paper proposes a method to characterize the decision boundary of deep neural networks. The authors take advantage of a new perspective on deep neural network i.e., tropical geometry. They present and prove a theorem connecting the decision boundary of deep neural networks to tropical hyperspheres.  Then, they use the theoretical results of the theorem for two applications i.e., network pruning and adversarial examples generation.  For the former, they show that using the introduced decision boundary characterization using the tropical geometry, one can dimish the number of parameters in a model with insignificant loss in performance. For the latter, they introduce a new method for adversarial examples generation by altering the decision boundary.

Overall, the technical and theoretical contribution of the paper regarding the relation of decision boundary to the tropical geometry is significant and could be useful for further investigation of the decision boundary of deep neural networks.
However, there are several caveats in this paper that need further clarification.

1) This paper needs to be placed properly among several important missing references on the decision boundary of deep neural networks [1][2]. In particular, using introduced tropical geometry perspective, how we can obtain the complexity of the decision boundary of a deep neural network?

2)  The second part of Theorem 2 should be explained straightforwardly and clearly as it plays an important role in the subsequent results and applications.

3) In the tropical network pruning section, the authors mention that "since fully connected layers in deep neural networks tend to have much higher memory complexity than convolutional layers, we restrict our focus to pruning fully connected layers". However, convolutional layers of investigated architectures (i.e., VGG16 and AlexNet) have a large number of parameters as well. So, wouldn't it be necessary to investigate pruning convolutional layers as well if diminishing the number of parameters is the main purpose? Moreover, the size of the last fully connected layer is simply determined by the preceding convolutional architecture, which in fact extracts the salient features (at least for well-known image datasets), while the last fully connected layer just flattens the extracted features to be fed into the subsequent classifier. Hence, it would be interesting and, in my opinion, necessary to investigate pruning convolutional layers as well.
Furthermore, the authors mention that a pruned subnetwork has a similar decision boundary to the original network. What does it mean exactly by "similar"? What are the measures capturing this similarity, if any? Also, the authors imply that two networks performing similarly (in terms of the accuracy) on a particular dataset have similar decision boundaries. I am skeptical if this the case and it needs to be validated concretely.

4) In adversarial examples generation, typically for a pre-trained deep neural network model one is interested in generating examples that are misclassified by the model while they resemble real instances. In this setting, we keep the model and thus its decision boundary intact. In this paper, nevertheless, aiming at generating adversarial examples, the decision boundary and thus the (pre-trained) model is altered. By chaining the decision boundary, however, the model's decisions for original real samples might change as well. Therefore, it is not clear to the reviewer how the introduced method is comparable to the well-established adversarial example generation setting.

5) Two previous papers investigated the decision boundary of the deep neural networks in the presence of adversarial examples [3][4]. Please discuss how the introduced method in this paper is placed among these methods.

Minor comments:
In the abstract, "We utilize this geometric characterization to shed light and new perspective on three tasks" --> unclear, needs to be revised.

Proposition 1, "the zonotope formed be the line segments" --> "the zonotope formed by the line segments"

Page 7. Results. "For VGG16, we perform similarly on both SVHN and CIFAR10 CIFAR100." --> "For VGG16, we perform similarly on both SVHN and CIFAR10."




References:
[1] @article{li2018decision,
  title={On the decision boundary of deep neural networks},
  author={Li, Yu and Richtarik, Peter and Ding, Lizhong and Gao, Xin},
  journal={arXiv preprint arXiv:1808.05385},
  year={2018}
}
[2] @article{beise2018decision,
  title={On decision regions of narrow deep neural networks},
  author={Beise, Hans-Peter and Da Cruz, Steve Dias and Schr{\"o}der, Udo},
  journal={arXiv preprint arXiv:1807.01194},
  year={2018}
}

[3] @article{khoury2018geometry,
  title={On the geometry of adversarial examples},
  author={Khoury, Marc and Hadfield-Menell, Dylan},
  journal={arXiv preprint arXiv:1811.00525},
  year={2018}
}

[4] @inproceedings{
he2018decision,
title={Decision Boundary Analysis of Adversarial Examples},
author={Warren He and Bo Li and Dawn Song},
booktitle={International Conference on Learning Representations},
year={2018},
url={https://openreview.net/forum?id=BkpiPMbA-},
}



**Experience Assessment:**

I have read many papers in this area.

**Review Assessment: Checking Correctness Of Derivations And Theory:**

I assessed the sensibility of the derivations and theory.

**Review Assessment: Checking Correctness Of Experiments:**

I assessed the sensibility of the experiments.

**Review Assessment: Thoroughness In Paper Reading:**

I read the paper at least twice and used my best judgement in assessing the paper.

---

> ### Author Response · Authors · 2019-11-12
> **Response (1/3) to R2**
>
> We thank R2 for the time spent reviewing the paper. We also thank R2 for acknowledging our technical and theoretical contributions. Please note that we have addressed the comments/typos/suggestions of all reviewers in the revised version and marked them in blue. Follows our response.
>
>
> Q1: This paper needs to be placed properly among several important missing references on the decision boundary of deep neural networks [1][2]. In particular, using introduced tropical geometry perspective, how we can obtain the complexity of the decision boundary of a deep neural network?
>
>
> The two works referenced by R2 are not directly related to the body of our work. Below, we summarize both works and state how our work is vastly different from both. The authors of [1] show that under certain assumptions, the decision boundaries of the last fully connected layer converges to an SVM classifier. That is to say, the features learnt in deep neural networks are linearly separable with max margin type linear classifier. On the other hand, the authors of [2] showed that the decision regions of neural networks with width smaller than the input dimension are unbounded. In our work, we use a new type of analysis (tropical geometry) to represent the set of decision boundaries $\mathcal{B}$ through its superset $\mathcal{T}(R(\mathbf{x}))$ that is the solution set to the tropical polynomial $R(\mathbf{x})$. We then show that this solution set is related to a geometric structure referred to as the decision boundaries polytope (convex hull between two zonotopes), this is analogous to constructing newton polytopes for the solution sets to classical polynomials in algebraic geoemtry. The normals to the edges of this polytope are parallel to the superset of the decision boundaries $\mathcal{T}(R(\mathbf{x}))$. That is to say, if one processes the polytope in an way that preserves the direction of the normals, the decision boundaries of the network are preserved. This is the base idea behind all later experiments. In general, this new representation presents a new fresh revisit to the lottery ticket hypothesis and an utterly new view to network pruning and adversarial attacks. We do believe this new representation can be of benefit to other applications and can open doors for a family of new geometrically inspired network regularizers as well.
>
> Q2: Regarding the complexity of the decision boundaries of neural networks.
>
> The only work at the intersection between TG and DNNs that we are aware of is the work of Zhang et. al. 2018. They re-derived a classical upper bound to the number of linear regions of feed forward neural networks with ReLU activations by counting vertices of polytopes (Theorem 6.3). The work was limited to the complexity (number of linear regions) of the functional representation of the network and not to the decision boundary.
>
>
>
> Q3: The second part of Theorem 2 should be explained straightforwardly and clearly as it plays an important role in the subsequent results and applications.
>
> We have added a "Digesting Theorem 2" paragraph in the revised version and rearranged the structure a bit around Theorem 2.
>
>
> Q4: Pruning convolutional layers.
>
> Most of the parameters (memory complexity) are the in fully connected layers. For example, the convolutional part of VGG16 has 14,714,688 parameters, whereas the fully connected layers have 262,000,400 parameters in total which is 17 times larger. Similarly, the convolutional part of AlexNet has 3,747,200 parameters while only the first fully connected layer has 37,752,832 parameters [5]. However, efficiently extending the tropical pruning to convolutional layers is a nontrivial interesting direction. Generally speaking, convolutional layers fit our framework naturally since a convolutional kernel can be represented with a structured topelitz/circulant matrix. However, a question of efficiency still remains as one still needs to construct the underlying structured matrix representing the convolutional kernel. Thereafter, a direction of interest is the tropical formulation of the network pruning problem as a function of the convolutional kernels surpassing the need for the construction of the dense representation of the kernel. We keep this for future work.

---

> > ### Author Response · Authors · 2019-11-12
> > **Response (2/3) to R2**
> >
> > Q5: For the similarity measure.
> >
> > Comparing the exact decision boundaries between two different architectures can be very difficult in the sense where decision boundaries for a two-class output network $f$ are defined as $\{\mathbf{x} \in \mathbb{R}^n : f_1(\mathbf{x}) = f_2(\mathbf{x})\}$. Another approach to compare decision boundaries, which is proposed by our work, is by computing the distance between the the dual subdivision polytope ($\delta(R(\mathbf{x}))$) of the tropical polynomials $R(\mathbf{x})$ representing two different architectures. This is since the normals to the edges of the polytope ($\delta(R(\mathbf{x}))$) are parallel to a superset of the decision boundaries (see Figure 1). This is exactly the proposed objective (1) where $d(.)$ is a distance function to compare the orientation between two general polytopes. Since finding a good choice for $d(.)$ is generally difficult we instead approximate it by comparing the generators constructing the ($\delta(R(\mathbf{x}))$) in Euclidean distance for ease (objective (2)). Experimentally and following prior art in the pruning literature (Han et. al. 2015), to compare the effectiveness of the pruning scheme, we compare the test accuracies across architectures as a function of the pruning ratio. Regardless, the reviewer is right about that similar test accuracies does not imply similar decision boundaries but rather only an indication.
> >
> >
> > Q6: In adversarial examples generation, typically for a pre-trained deep neural network model one is interested in generating examples that are misclassified by the model while they resemble real instances. In this setting, we keep the model and thus its decision boundary intact. In this paper, nevertheless, aiming at generating adversarial examples, the decision boundary and thus the (pre-trained) model is altered. By chaining the decision boundary, however, the model's decisions for original real samples might change as well. Therefore, it is not clear to the reviewer how the introduced method is comparable to the well-established adversarial example generation setting.
> >
> > The new approach is definitely comparable to the well-establisehd adversarial example generation setting. Let us explain. The new analysis provided by Theorem 2, allows to present the decision boundaries geometrically as a convex hull between two zonotopes which is a function of only the network parameters and not the input space. Thus, it not clear how one can frame the adversarial attacks problem in this new fresh tropical setting since adversarial attacks is the task of perturbing the input space as opposed to the parameters space of the network resulting in a flip in the prediction. In the tropical adversarial attacks section, we show that the problem of designing an adversarial attack $\mathbf{x}_0 + \eta$ that flips the network prediction is closely related to the problem of flipping the network prediction by perturbing the network parameters in the first layer $\mathbf{A}_1 + \zeta_{\mathbf{A}_1}$ where both problems are related through a linear system. That is to say, if one finds $\zeta_{\mathbf{A}_1}$ (perturbations in the first linear layer) that perturbs the geometric structure (convex hull between two zonotopes, i.e. decision boundaries) sufficiently enough to flip the network prediction, one can find an equivalent pixel adversarial attack $\eta$ by solving the linear system $\mathbf{A}_1 \eta = \zeta_{\mathbf{A}_1}\mathbf{x}_0$ that flips the prediction of the original unperturbed network (see the end of page 7). We incorporate this in an overall objective (5) where the linear system is incorporated as a constraint. Upon solving (5) with the proposed Algorithm 1, we attack the original network (unperturbed) with the adversarial attack $\eta$. Therefore, this is comparable to the classical adversarial attacks framework. This approach indeed resulted into flipping the network prediction over all tested examples on the MNIST dataset. As highlighted at the end of section 6, we do not aim in our approach to outperform the state of the art adversarial attacks, but rather to provide a novel geometrically inspired perspective that can shed new light in this field.

---

> > > ### Author Response · Authors · 2019-11-12
> > > **Response (3/3) to R2**
> > >
> > > Q7: Two previous papers investigated the decision boundary of the deep neural networks in the presence of adversarial examples [3][4]. Please discuss how the introduced method in this paper is placed among these methods.
> > >
> > > The work of [3] analyzed the geometry of adversarial examples by means of manifold reconstruction to study the trade off between robustness under different norms. On the other hand, [4] crafted adversarial attacks by estimating the distance to the decision boundaries using random search directions. Both of the papers made a local estimation of the decision boundary around the attacked point to construct the adversarial attack to the input image. In our work, we geometrically characterized the decision boundaries in Theorem 2 where the polytope (convex hull of two zonotopes) is only a function of the network parameters and NOT the input space. We presented a dual view to adversarial attacks in which one can construct adversarial examples by investigating network parameters perturbations that results in the largest perturbation to this polytope representing the decision boundaries. The scope of our work, and unlike prior art [3,4], is focused towards a new geometric polytope representation of the decision boundary in the network parameter space (not the input space) through a new novel analysis. We have added a discussion of both papers in the adversarial attacks section (Section 6).
> > >
> > >
> > >
> > > Minor comments.
> > >
> > > We have addressed the concerns of R2 and left the changes in blue in the revised version.
> > >
> > >
> > > [1] "On the decision boundary of deep neural networks". SLi, Yu and Richtarik, Peter and Ding, Lizhong and Gao, Xin.
> > >
> > > [2] "On decision regions of narrow deep neural networks". Beise, Hans-Peter and Da Cruz, Steve Dias and Schr{\"o}der, Udo.
> > >
> > > [3] "On the geometry of adversarial examples". Marc Khoury and  and Dylan Hadfield-Menell.
> > >
> > > [4] "Decision Boundary Analysis of Adversarial Examples".  Warren He, Bo Li and Dawn Song.
> > >
> > > [5] "Learning both weights and connections for efficient neural networks". Song Han, Jeff Pool, John Tran, and William J. Dally.

---

### Official Review · AnonReviewer3 · 2019-10-23
**Official Blind Review #3**

**Rating:** 1

**Review:**

This paper proposed a framework based on a mathematical tool of tropical geometry to characterize the decision boundary of neural networks. The analysis is applied to network pruning, lottery ticket hypothesis and adversarial attacks.

I have some questions:

Q1: What benefit does introducing tropical geometry brings in terms of theoretical analysis? Does using tropical geometry give us the theoretical results that traditional analysis can not give us? If so, what is it? I am trying to understand why the authors use this tool. The authors should be explicit in their motivation so that the readers are clear about the contribution of this paper. More specifically, from my perspective, tropical semiring, tropical polynomials and tropical rational functions all can be represented with the standard mathematical tools. Here they are just redefining several concepts.

Q2: In “Experiments on Tropical Pruning”, the authors mentioned “we compare our tropical pruning approach against Class Blind (CB), Class Uniform (CU), and Class Distribution (CD) methods Han et al. (2015)”. What is Class Blind, Class Uniform and Class Distribution? There seems to be an error here “Figure 5 shows the pruning comparison between our tropical approach ...”, i think Figure 5 should be Figure 4.

Q3: In the adversarial attack part, is the authors proposing a new attack method? If so, then the authors should report the test accuracy under attack. Also, the experimental results should not be restricted to MNIST dataset. I am also not sure about the attack settings here, the authors said “Instead of designing a sample noise η such that (x0 + η) belongs to a new decision region, one can instead fix x0 and perturb the network parameters to move the decision boundaries in a way that x0 appears in a new classification region.”. Why use this setting? Are there any intuitions? Since this is different from traditional adversarial attack terminology, the authors should stop using adversarial attacks as in “tropical adversarial attacks” because it is really misleading.



===================================================================
Thanks the authors for the response. I still have two questions:

Q1: The authors say that this theory provides a deeper understanding to Lottery Ticket Hypothesis (LTH). Then another paper “Rethinking the Value of Network Pruning” [1] suggests something different than LTH. [1] suggests that we do not need the initialization of large networks to train the pruned network from scratch to achieve high accuracy. Since the authors claim that their theory is related to LTH, then what would the proposed theory say about [1]?

Q2: Since you redesign the task of adversarial attacks, I am still not convinced why this setting is interesting? The reason why people are interested in adversarial attacks is because it could happen during test time. What is the application of this setting? Why this new setting is important and worth studying? They are not clear to me.
Also, as i wrote in my initial review, “the authors should stop using “adversarial attacks” as in “tropical adversarial attacks” because it is really misleading.”. I hope the authors can address this concern, or otherwise new readers may also find this part difficult to understand.

[1] Rethinking the Value of Network Pruning. Zhuang Liu, Mingjie Sun, Tinghui Zhou, Gao Huang, Trevor Darrell. ICLR 2019.



**Experience Assessment:**

I have read many papers in this area.

**Review Assessment: Checking Correctness Of Derivations And Theory:**

I carefully checked the derivations and theory.

**Review Assessment: Checking Correctness Of Experiments:**

I assessed the sensibility of the experiments.

**Review Assessment: Thoroughness In Paper Reading:**

I read the paper at least twice and used my best judgement in assessing the paper.

---

> ### Author Response · Authors · 2019-11-12
> **Response (1/2) to R3**
>
> We thank R3 for the time spent reviewing the paper. It is though not clear to the authors the main reason behind the initial score of weak reject as R3 seems to have very generic questions about our work but not a particular criticism of the novelty/contribution that we can address in our rebuttal. We hope that the following response addresses and clarifies some key elements. Moreover, we want to bring to the attention of R3 that we have addressed the comments/typos/suggestions of all reviewers in the revised version and marked them in blue.
>
>
> Q1: What benefit does introducing tropical geometry brings in terms of theoretical analysis? Does using tropical geometry give us the theoretical results that traditional analysis can not give us? If so, what is it? I am trying to understand why the authors use this tool. The authors should be explicit in their motivation so that the readers are clear about the contribution of this paper. More specifically, from my perspective, tropical semiring, tropical polynomials and tropical rational functions all can be represented with the standard mathematical tools. Here they are just redefining several concepts.
>
>
> As discussed thoroughly in the introduction (last paragraph of page 1), tropical geometry is the younger twin to algebraic geometry on a particular semiring defined in a way to align with the study of piecewise linear functions. The early definitions stated in the paper (1 to 5) are well known in the TG literature and were restated for the completion of this paper. While it is true that the definitions can be represented with standard mathematical tools; however, this misses the fundamental powerful element TG promises. TG transforms algebraic problems of piecewise linear nature to a combinatoric problem on general polytopes. To that end, Zhang et. al. 2018 (to the best of our knowledge the only work at the intersection between TG and DNNs) rederived classical results (upper bound on the number of linear pieces of DNNs) in a much simpler analysis by counting vertices on polytopes. In this work, instead of studying the functional representation of piecewise linear DNNs, we study their decision boundaries using the lens of TG. To wit, the geometric characterization of the decision boundaries of DNNs developed in Theorem 2 cannot be attained using standard mathematical tools. More specifically, Theorem 2 represented a superset to the decision boundaries (the tropical hypersurface $\mathcal{T}(R(\mathbf{x}))$, with a geometric structure that is the convex hull between two zonotopes. While this by itself opens doors for a family of new geometrically motivated regualrizers for training DNNs that are in direct correspondence with the behaviour of the decision boundaries, we do not dwell on training beyond this point and leave that for future work. However, this new result allowed for
> re-affirmation to the lottery ticket hypothesis in a new fresh perspective. Moreover, we propose new optimization problems that are geometrically motivated (based on Theorem 2) for several classical problems, i.e. network pruning and adversarial attacks that were not possible before and have provided several new insights and directions. That is we show an intimate relation between network perturbations (through decision boundary polytope perturbations) and the construction of adversarial attacks (input perturbations).
>
> Q2: In “Experiments on Tropical Pruning”, the authors mentioned “we compare our tropical pruning approach against Class Blind (CB), Class Uniform (CU), and Class Distribution (CD) methods Han et al. (2015)”. What is Class Blind, Class Uniform and Class Distribution? There seems to be an error here “Figure 5 shows the pruning comparison between our tropical approach ...”, i think Figure 5 should be Figure 4.
>
>
> We have added the definition of the pruning methods of [1] in the revised version of the paper for completion,  and corrected the typo in the Figure reference.
>
>
> [1] "Learning both weights and connections for efficient neural networks". Song Han, Jeff Pool, John Tran, and William J. Dally.

---

> > ### Author Response · Authors · 2019-11-12
> > **Response (2/2) to R3**
> >
> > Q3: In the adversarial attack part, is the authors proposing a new attack method? If so, then the authors should report the test accuracy under attack. Also, the experimental results should not be restricted to MNIST dataset. I am also not sure about the attack settings here, the authors said “Instead of designing a sample noise η such that (x0 + $\eta$) belongs to a new decision region, one can instead fix x0 and perturb the network parameters to move the decision boundaries in a way that x0 appears in a new classification region.”. Why use this setting? Are there any intuitions? Since this is different from traditional adversarial attack terminology, the authors should stop using adversarial attacks as in “tropical adversarial attacks” because it is really misleading.
> >
> >
> > As highlighted in the last sentence in section 6, we are not competing against other attacks, but we rather show how this new geometric view to the decision boundaries provided by the TG analysis in Theorem 2 can be leveraged for the construction of adversarial attacks. We want to emphasize to R3 that the polytope representing the decision boundaries (convex hull of two zonotopes as per Theorem 2) is a function of the network parameters and not of the input space. Thus, it is not initially clear how one can frame the adversarial attacks problem in this new fresh tropical setting since adversarial attacks is the task of perturbing the input space as opposed to the parameters space of the network resulting in a flip in the prediction. In the tropical adversarial attacks section, we show that the problem of designing an adversarial attack $\mathbf{x}_0 + \eta$ that flips the network prediction is closely related to the problem of flipping the network prediction by perturbing the network parameters in the first layer $\mathbf{A}_1 + \zeta_{\mathbf{A}_1}$ where both problems are related through a linear system. That is to say, if one finds $\zeta_{\mathbf{A}_1}$ that perturbs the geometric structure (convex hull between two zonotopes, i.e. decision boundaries) sufficiently enough to flip the network prediction, one can find an equivalent pixel adversarial attack $\eta$ by solving the linear system $\mathbf{A}_1 \eta = \zeta_{\mathbf{A}_1}\mathbf{x}_0$ that flips the prediction of the original unperturbed network (see the end of page 7). We thereafter propose Problem (5) incorporating the geometric information from Theorem 2 where the linear system is accounted for in the constraints set. We propose an algorithm to solve the problem (a mix of penalty and ADMM) detailed in Algorithm 1 in the appendix. The solution to problem 5 by applying Algorithm 1 results in the construction of adversarial attacks ($\eta$) that indeed flip the network prediction over all tested examples on the MNIST dataset.

---

### Author Response · Authors · 2019-09-29
**Link to code**

We would like to bring to the attention of the reviewers that we have disabled the link sharing to the code since, while it is very unlikely unless purposely looked for, the affiliation of the authors may be exposed.

To that regard, we attach the code here. The new link is

https://drive.google.com/file/d/1IpJYa7RVzlfgg_uVi-I-NZoL94PeM0kl/view

---

### Decision · Program_Chairs · 2019-12-19

**Decision:**

Reject

**Comment:**

This paper studies the decision boundaries of a certain class of
neural networks (piecewise linear, non-linear activation functions)
using tropical geometry, a subfield of algebraic geometry that leverages piece-wise linear structures.
Building on earlier work, such piecewise linear networks are shown to be represented as
a tropical rational function. This characterisation is used to explain different phenomena of neural
network training, such as the 'lottery ticket hypothesis', network
pruning, and adversarial attacks.

This paper received mixed reviews, owing to its very specialized area. Whereas R1 championed the submission
for its technical novelty, the other reviewers felt the current exposition is too inaccessible and some application areas are not properly addressed. The AC shares these concerns, recommends rejection and strongly encourages the authors to address the reviewers concerns in the next iteration.